Corrected: Author correction

# Time-dependent memory transformation along the hippocampal anterior–posterior axis

Lisa C. Dandolo[1] & Lars Schwabe[1]

With time, memories undergo a neural reorganization that is linked to a transformation of detailed, episodic into more semantic, gist-like memory. Traditionally, this reorganization is thought to involve a redistribution of memory from the hippocampus to neocortical areas. Here we report a time-dependent reorganization within the hippocampus, along its anterior–posterior axis, that is related to the transformation of detailed memories into gist-like representations. We show that mnemonic representations in the anterior hippocampus are highly distinct and that anterior hippocampal activity is associated with detailed memory but decreases over time. Posterior hippocampal representations, however, are more gist-like at a later retention interval, and do not decline over time. These findings indicate that, in addition to the well-known systems consolidation from hippocampus to neocortex, there are changes within the hippocampus that are crucial for the temporal dynamics of memory.

[1] Department of Cognitive Psychology, University of Hamburg, 20146 Hamburg, Germany. Correspondence and requests for materials should be addressed to L.S. (email: Lars.Schwabe@uni-hamburg.de)

Memories evolve over time. After initial encoding, new information becomes fixed at a cellular level and integrated within networks of existing memories[1,2]. This integration involves a reorganization of memory during which, with time, detailed, episodic memories are transformed into more semantic, gist-like representations[1,3]. Although, the neural underpinnings of this time-dependent memory reorganization are at the heart of the neuroscience of memory, the neural evolution of memories over time remains a topic of much controversy. In particular, whether the hippocampus, a critical hub for initial memory formation[4–8], is involved in remote memories or not has been controversial for decades[3,9–12].

The hippocampus can be subdivided into anterior and posterior parts—corresponding to the ventral and dorsal hippocampus, respectively, in rodents—and these parts differ in function, structure and their connections to cortical and subcortical areas[13–15]. A prominent proposal that was largely based on rodent data linked the ventral (anterior) hippocampus to emotion, stress, and affect, whereas the dorsal (posterior)

hippocampus was implicated in cognitive functions such as learning, memory, and spatial navigation[16]. Electrophysiological and lesion studies in rodents, as well as human neuroimaging studies, however, suggest that this view may need to be revised and that both anterior and posterior hippocampal areas (aHC and pHC, respectively) may contribute to learning and memory processes, although the exact functional specialization is still unclear[14,15,17]. Further studies suggest that the aHC and pHC might be differentially involved in recent and remote memories[18–20], yet whether the transformation of memory over time may be linked to time-dependent changes in aHC and pHC involvement in memory is completely unknown.

Here we determine whether there are time-dependent changes in aHC and pHC contributions to memory and if so, whether they are associated with the transformation from detailed to gist-like memory. To do so, we combined functional magnetic resonance imaging (fMRI) and multivariate representational similarity analysis (RSA) with a task probing memory transformation. Participants learned 60 pictures of scenes and objects (30 neutral,

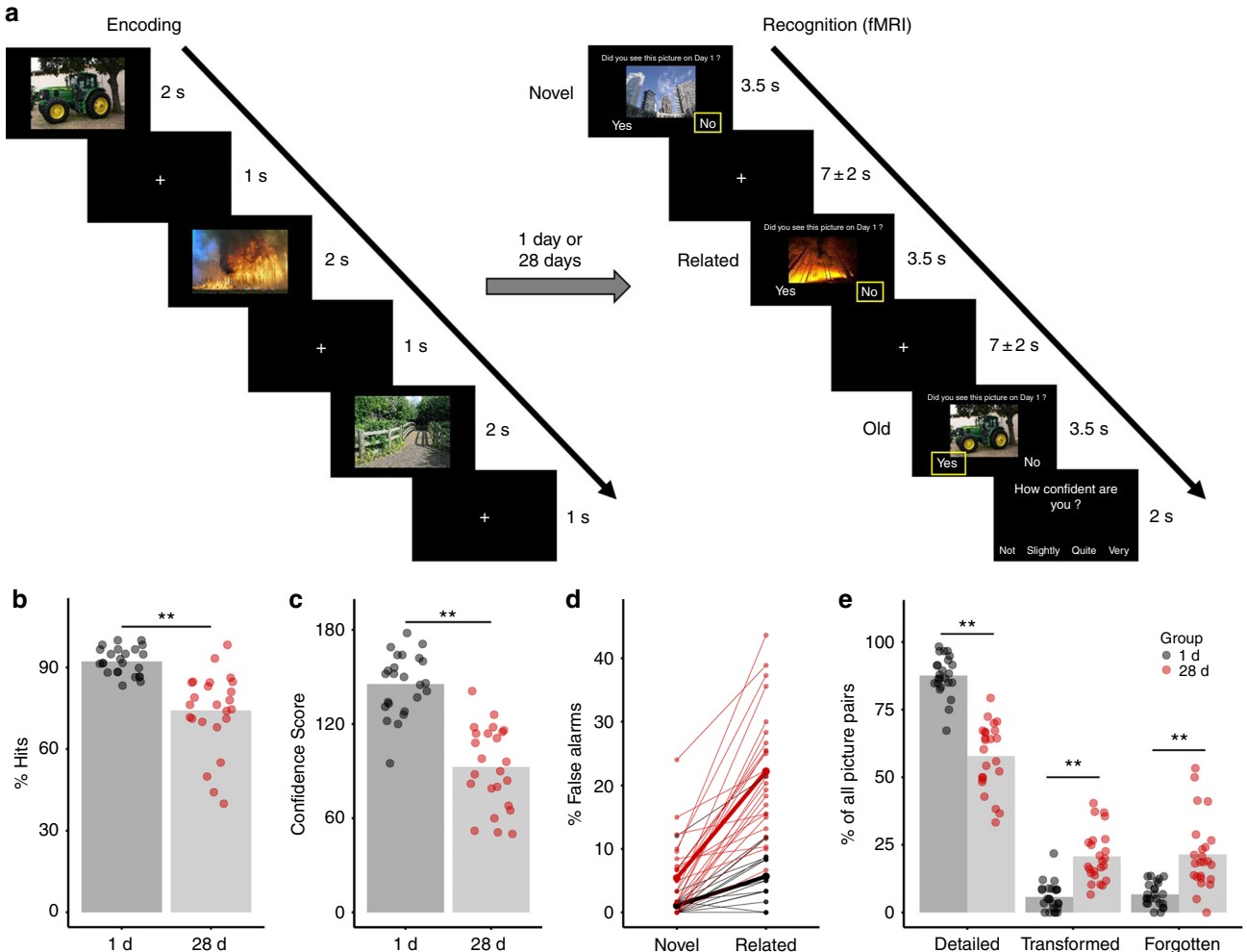

**Fig. 1** Task and behavioral results. **a** Schematic overview of the picture encoding task (experimental day 1) and recognition task (experimental day 2). Images courtesy of Andreas Praefcke (tractor), Morio (skyscrapers), USDA (wildfires), and Acabashi (footbridge). **b** Percentage of Hits: participants in the 28 d group showed significantly less hits (main effect Group: $F_{(1,46)} = 33.57$, $p = 5.89e-07$, generalized $\eta^2 = 0.377$, $n = 48$). **c** Confidence Score: participants in the 28 d group had a significantly lower Confidence Score (main effect Group: $F_{(1,46)} = 63.33$, $p = 3.41e-10$, generalized $\eta^2 = 0.550$, $n = 48$). **d** Percentage of FA for related pictures and novel pictures: the increase in FA from the 1 d group to the 28 d group was more pronounced for related pictures than novel pictures (Picture Type × Group interaction: $F_{(1,46)} = 36.31$, $p = 2.65e-07$, generalized $\eta^2 = 0.155$, $n = 48$). **e** Percentage of picture pairs: the 28 d group compared to the 1 d group showed fewer detailed pairs (main effect Group: $F_{(1,46)} = 102.96$, $p = 2.55e-13$, generalized $\eta^2 = 0.607$, $n = 48$), more forgotten pairs (main effect Group: $F_{(1,46)} = 26.88$, $p = 4.72e-06$, generalized $\eta^2 = 0.335$, $n = 48$) and critically also more transformed pairs (main effect Group: $F_{(1,46)} = 45.15$, $p = 2.41e-08$, generalized $\eta^2 = 0.425$, $n = 48$). **p < 0.001

30 negative) and performed a recognition test for these pictures in the MRI scanner either one day after encoding (1 d group) or four weeks later (28 d group). Critically, the recognition test included, in addition to the old pictures learned during encoding and completely novel pictures, related lure pictures that carried the semantic gist of the old pictures but had different details (Fig. 1a).

The endorsement of related pictures as "old" provided a behavioral index of the time-dependent transformation from detailed to more gist-like memory representations.

As predicted, we found a strong increase in the endorsement of related pictures as old in the 28 d group relative to the 1 d group. This finding indicates a time-dependent memory transformation.

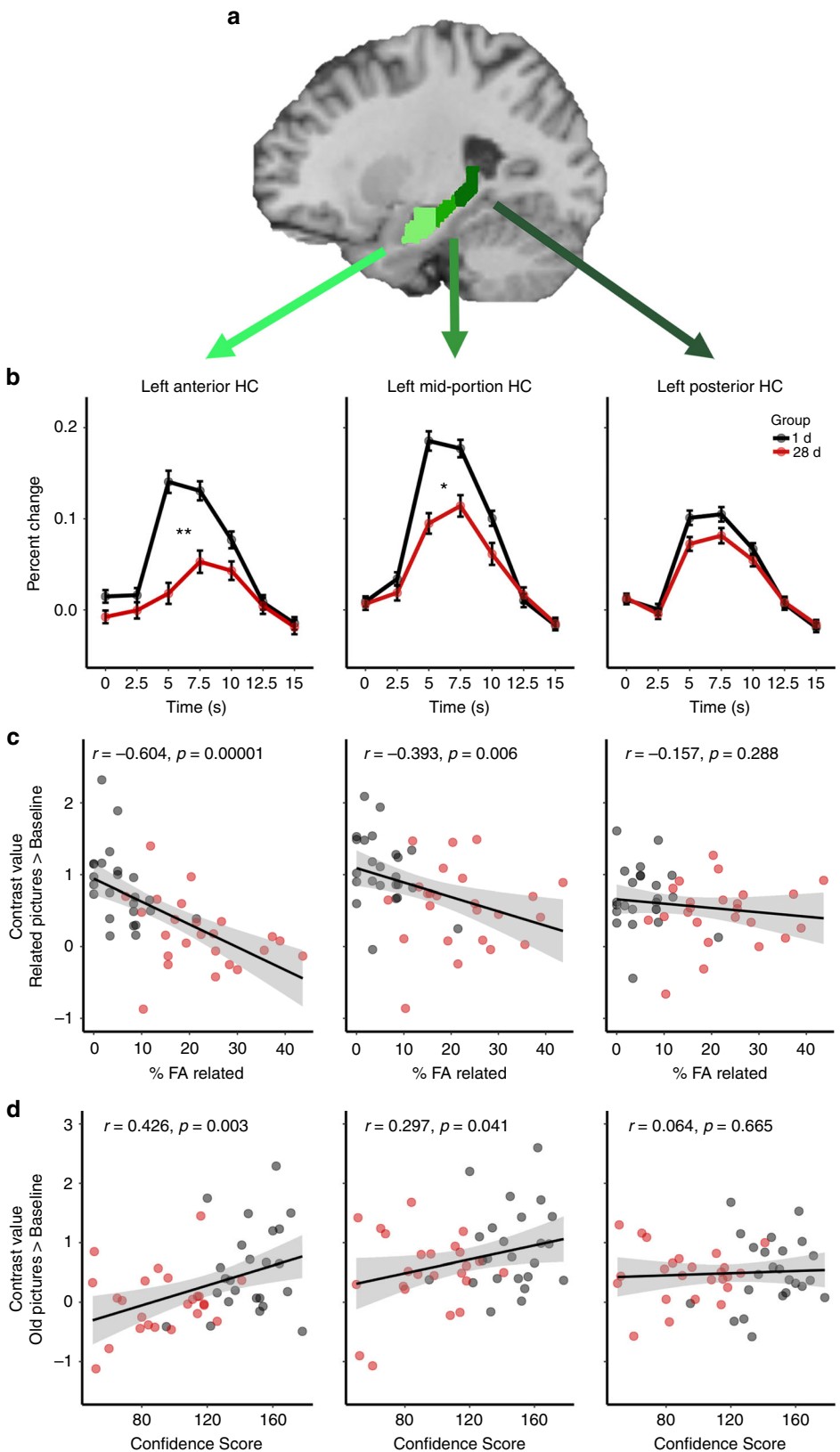

At the neural level, this transformation was paralleled by a time-dependent decrease in the aHC, the hippocampal subregion that was directly linked to memory specificity. The pHC, in turn, was not related to memory specificity and did not decline over time. Further, RSA revealed that activity patterns in the aHC were highly specific and differed between old and new memories in the 1 d group but not after 28 d. Representations in the pHC became more gist-like after 28 days. Together, these findings show that the aHC that supports memory specificity declines over time, whereas pHC remains stable over time but carries more gist-like representations. Our data suggest that the time-dependent transformation from detailed to gist-like memory is linked to a reorganization within the hippocampus.

## Results

**Time-dependent memory transformation.** During encoding on experimental day 1, participants of the 1 d- and 28 d groups learned the pictures equally well (Supplementary Fig. 1). In the recognition test, either 1 day or 28 days after encoding, the hit rate was expectedly lower in the 28 d group than in the 1 d group (main effect Group: $F_{(1,46)} = 33.57$, $p = 5.89e{-}07$, generalized $\eta^2 = 0.377$, $n = 48$; Fig. 1b), yet memory was still clearly intact after 28 d as reflected by a hit rate of ~75% and a false alarm (FA) rate for novel lures of only 5%. Most importantly, however, participants in the 28 d group showed a sharp increase in the FA rate specifically for related pictures and to a significantly lesser extent for novel pictures (Picture Type × Group interaction: $F_{(1,46)} = 36.31$, $p = 2.65e{-}07$, generalized $\eta^2 = 0.155$, $n = 48$; main effect Group: $F_{(1,46)} = 45.63$, $p = 2.13e{-}08$, generalized $\eta^2 = 0.343$, $n = 48$; main effect Picture Type: $F_{(1,46)} = 113.18$, $p = 5.48e{-}14$, generalized $\eta^2 = 0.363$, $n = 48$; Fig. 1d). For related pictures, the FA rate rose to almost 25% after 28 days and was thus more than four times higher than the FA rate for novel pictures. This indicates that participants in the 28 d group particularly had difficulties differentiating between old pictures and related pictures carrying the gist of the old pictures, suggesting a transformation towards more gist-like memory. Additionally, participants were asked to rate their confidence on a 4-point-scale, whenever they indicated that they had seen the picture before (Fig. 1a), allowing us to calculate a Confidence Score as another measure of memory specificity. Participants in the 28 d group were, as expected, significantly less confident in their memory than participants in the 1 d group (main effect Group: $F_{(1,46)} = 63.33$, $p = 3.41e{-}10$, generalized $\eta^2 = 0.550$, $n = 48$; Fig. 1c).

We further analyzed the 60 matching picture pairs (i.e., old pictures learned during encoding and their respective related lures) and categorized memories for them as being either detailed, transformed or forgotten depending on whether participants endorsed solely the old pictures, both the old and related pictures,

or none of them as "old". Participants in the 28 d group showed, compared to those of the 1 d group, significantly fewer detailed (main effect Group: $F_{(1,46)} = 102.96$, $p = 2.55e{-}13$, generalized $\eta^2 = 0.607$, $n = 48$) and more forgotten memories (main effect Group: $F_{(1,46)} = 26.88$, $p = 4.72e{-}06$, generalized $\eta^2 = 0.335$, $n = 48$), but critically also more transformed memories (main effect Group: $F_{(1,46)} = 45.15$, $p = 2.41e{-}08$, generalized $\eta^2 = 0.425$, $n = 48$; Fig. 1e), again in line with the proposed time-dependent memory transformation. Our behavioral data further suggest that stimulus-related emotional arousal influenced the transformation to gist-like memories: after 28 days significantly fewer negative pictures were forgotten than neutral ones (paired t-test: $t(23) = -5.00$, $p = 4.64e{-}05$, Cohen's $d = -1.02$, $n = 24$) and, even more interestingly, negative pictures were significantly more often transformed than neutral ones (paired t-test: $t(23) = 2.67$, $p = 0.0138$, Cohen's $d = 0.54$, $n = 24$; Supplementary Fig. 2), in line with findings[21,22] suggesting that superior memory for emotional material, indicated here by the slower forgetting rate, comes at the cost of reduced memory for contextual details, reflected here in an increase in transformed memories.

**aHC but not pHC activity decreases over time.** To elucidate the neural underpinnings of the memory dynamics over time, we first analyzed time-dependent changes in the hippocampus as a whole and other cortical and subcortical areas that have been implicated in episodic memory before. We obtained overall reduced activity in the hippocampus, parahippocampus, and the amygdala in the 28 d group compared to 1 d group (see Supplementary Fig. 3 and Supplementary Tables 1 and 2). In addition, we performed a psychophysiological interaction (PPI) analysis to test whether the cross-talk of the hippocampus with other areas critical for memory formation changed as a function of time. Our analysis showed specifically reduced functional connectivity between the right hippocampus and the right amygdala in the 28 d group compared to the 1 d group (Supplementary Fig. 4). This decrease in hippocampal-amygdala connectivity was of particular interest as the interaction of these areas is commonly linked to vivid memory[23].

As hippocampal involvement in remote memories has been argued to depend critically on memory vividness[3,20], we further explicitly looked at activity for old items that were recognized with high confidence. Even for those High Confidence Hits, overall hippocampal activity was lower in the 28 d group than in the 1 d group (Supplementary Fig. 5a). For neocortical areas involved in more semantic or schema-related memory processes, there was, however, no reliable difference in activity in the 28 d group vs. the 1 d group (when correcting for the number of ROIs; Supplementary Table 2). As we tested memory with a recognition test in which participants directly viewed all pictures, it may not

**Fig. 2** Univariate analysis of the left HC long axis ROIs. **a** Depiction of the three hippocampal ROIs: aHC ($Y = -4$ to $-18$) = light green, mHC ($Y = -19$ to $-29$) = green, pHC ($Y = -30$ to $-40$) = dark green. Visualizations of the anatomical masks are superimposed on a sagittal section of a template image. **b** FIR time courses over the first 15 s (7TRs) for all picture types combined. Statistical comparisons were calculated for the peak response (average of the 5 s and 7.5 s time points). The activity in the 28 d group compared to the 1 d group decreased in the aHC (main effect Group: $F_{(1,46)} = 15.46$, $p = 0.0003$, generalized $\eta^2 = 0.1546$, $n = 48$) and mHC ($F_{(1,46)} = 9.24$, $p = 0.0038$, generalized $\eta^2 = 0.0999$, $n = 48$). There was no difference between the groups in the pHC ($F(1,46) = 1.70$, $p = 0.1986$, generalized $\eta^2 = 0.0230$, $n = 48$). For effects of Picture Type and Emotion see Supplementary Fig. 6. **c** Correlations of the percentage of FA to related items with the contrast value for related pictures vs baseline: across groups, there was a negative correlation in the aHC and mHC, but not in the pHC. Calculating the correlations separately for each group showed a significant correlation in the aHC for the 1 d group ($t(22) = -2.37$, $p = 0.027$, Pearson's $r = -0.45$, $n = 24$) and a trend in the 28 d group ($t(22) = -1.62$, $p = 0.121$, Pearson's $r = -0.37$, $n = 24$), while correlations for each group separately were not significant in the mHC and pHC. **d** Correlations of the Confidence Score with the contrast value for old pictures vs baseline: across groups, there was a positive correlation in the aHC and mHC, but not in the pHC. Note, however, that none of the correlations with the Confidence Score were significant when calculating them separately for each group. For analysis of the right HC long axis ROIs see Supplementary Fig. 7. *$p < 0.05$, **$p < 0.001$, all error bars are SEM

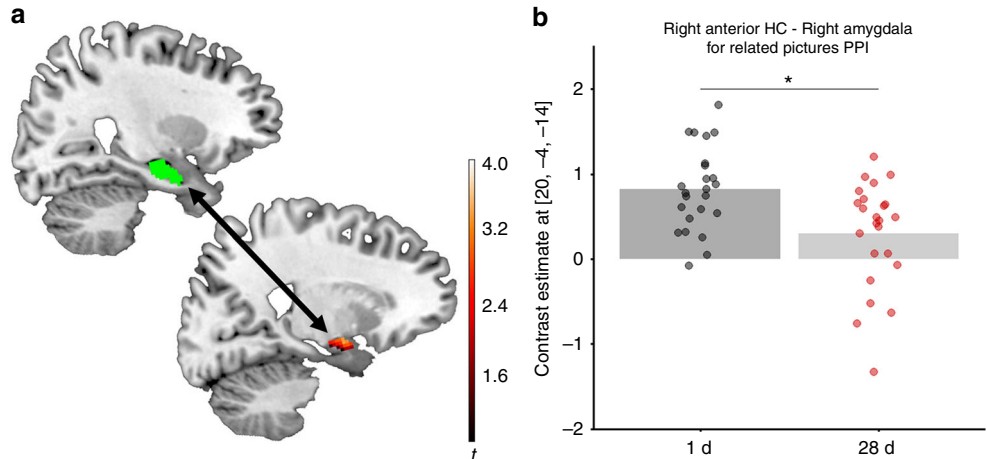

**Fig. 3** Connectivity with amygdala. **a** Visualization of the connectivity between right aHC and right amygdala. Green represents the anatomical right aHC mask; red represents activation in the right amygdala for the contrast 1 d > 28 d of PPI interactions for related pictures with the right aHC as seed. Visualizations are superimposed on sagittal sections of a T1-weighted template image. **b** Parameter estimates in the peak voxel in the right amygdala. The right aHC showed a reduced connectivity to the right amygdala in the 28 d group compared to the 1 d group for related pictures (SVC peak level: $x = 20$, $y = -4$, $z = -14$, $t = 3.59$, $p$(FWE) $= 0.0156$, $k = 15$), but not for old or novel pictures. Note that neither using the left aHC nor the pHC (left or right) as seed regions, nor looking at other ROIs resulted in significant group differences in connectivity. See also Supplementary Table 3. *$p < 0.05$

be surprising that neocortical areas were similarly involved in the 1 d- and 28 d-group, as these areas might just reflect the processing of the currently viewed pictures. The fMRI findings so far are generally in line with the systems consolidation view, which would predict a decrease of hippocampal involvement in memory, irrespective of the specific picture type, over time[9,12,24].

Looking at the hippocampal subregions along the long axis (Fig. 2a), however, revealed that the time-dependent decrease in activity was restricted to the aHC (left aHC main effect Group: $F_{(1,46)} = 15.46$, $p = 0.0003$, generalized $\eta^2 = 0.1546$, $n = 48$) and mid-portion hippocampus (left mHC: $F_{(1,46)} = 9.24$, $p = 0.0038$, generalized $\eta^2 = 0.0999$, $n = 48$). Activity in the pHC, however, did not significantly differ between the 1 d- and 28 d-group (left pHC: $F_{(1,46)} = 1.70$, $p = 0.1986$, generalized $\eta^2 = 0.0230$, $n = 48$; Fig. 2b). These differences between the ROIs were underlined by a significant Group × HC Long Axis interaction ($F(2,92) = 6.07$, $p = 0.0033$, generalized $\eta^2 = 0.036$, $n = 48$). The decrease in activity for high-confidence hits was also most pronounced in the aHC (Supplementary Fig. 5b). Moreover, connectivity analysis using the aHC and pHC as seed regions showed that the right aHC-right amygdala connectivity for related pictures (but not old or novel pictures) was significantly reduced in the 28 d- relative to the 1 d-group (SVC peak level: $x = 20$, $y = -4$, $z = -14$, $t = 3.59$, $p$(FWE) $= 0.0156$, $k = 15$; Fig. 3), while we found no significant differences between the groups in the connectivity to the amygdala when using the pHC as seed region, suggesting that it might be the connectivity between the aHC and the amygdala that is notably reduced in the 28 d group.

In order to further examine whether the decrease in aHC activity could be directly linked to the change in the nature of remembering, we correlated the activity in the hippocampal subregions with behavioral indices of memory specificity, i.e., the FA rate for related lures and the Confidence Score. These analyses showed that specifically the aHC was associated with the specificity of memory. In particular, aHC activity for related pictures was correlated negatively with the FA rate to related pictures (left aHC: $t_{(46)} = -5.15$, $p = 5.37e-06$, Pearson's $r = -0.60$, $n = 48$; Fig. 2c) and aHC activity for old pictures correlated positively with the Confidence Score (left aHC: $t(46) = 3.19$, $p = 0.0025$, Pearson's $r = 0.43$, $n = 48$; Fig. 2d). For the pHC, however, there were no such associations with memory

specificity (left pHC, FA rate to related pictures: $t(46) = -1.08$, $p = 0.2879$, Pearson's $r = -0.16$, $n = 48$; Confidence Score: $t(46) = 0.44$, $p = 0.6645$, Pearson's $r = 0.06$, $n = 48$) and the correlations between activity and indicators of memory specificity were significantly distinct in the left aHC and pHC (FA related: Pearson and Filon's $z = -3.46$, $p = 0.0005$, $n = 48$; Confidence Score: Pearson and Filon's $z = 2.81$, $p = 0.0049$, $n = 48$). These correlations across the 1 d- and 28 d-groups indicate that the aHC and pHC are differentially linked to memory specificity. When we looked at the correlations separately in the 1 d- and 28 d-group, we obtained for the percentage of FA to related items, the key parameter of memory specificity, a significant correlation with aHC in the 1 d group only ($t(22) = -2.37$, $p = 0.027$, Pearson's $r = -0.45$, $n = 24$). For the 28 d group, this correlation did not reach significance ($t(22) = -1.62$, $p = 0.121$, Pearson's $r = -0.37$, $n = 24$), which might be related to the proposed reduced involvement of the aHC in memory in the 28 d group, although a lack of statistical power might also account for the non-significant correlation in the 28 d group. For the memory Confidence Score, the correlations with aHC activity did not reach significance in the separate groups (1 d group: $t(22) = 1.17$, $p = 0.255$, Pearson's $r = 0.24$, $n = 24$; 28 d group: $t(22) = 1.01$, $p = 0.326$, Pearson's $r = 0.21$, $n = 24$).

Although, we found a reduction of activity in the 28 d group in comparison to the 1 d group in the aHC and the mHC, it is important to note that there was no Group × Picture Type interaction (Supplementary Fig. 6a) in these ROIs. Thus, our univariate results show that the activity in the aHC, but not the pHC, is reduced in the 28 d group compared to the 1 d group for all picture types, suggesting that the contribution of the aHC in the task in general is reduced. Our brain-behavior correlations further show that the aHC, but not the pHC, is associated with memory specificity. It is not surprising that aHC activity was reduced irrespective of Picture Type after 28 d because a specific memory representation is required to both correctly identify an old item as old and to correctly reject novel or related items.

**Specificity of mnemonic representations in aHC and pHC.** The above univariate analyses showed that it was specifically the aHC that was associated with memory specificity and that specifically

activity in this area decreased at a longer retention interval of 28 days. While univariate analyses can show a general involvement of an area in a task, multivariate analysis allows the detection of specific patterns of activity across multiple voxels and may be more sensitive to the changing representations of the different picture types and more informative about the functional organization of memory at different time intervals. Therefore, we

ran a RSA (Fig. 4a) to examine whether the mnemonic representations differed in the aHC, mHC and pHC, whether they changed depending on the retention interval, and to what extent such different representational patterns can be linked to the proposed memory transformation. We first created average representational dissimilarity matrices (RDMs) in each hippocampal subregion, separately for each group (Fig. 4b):

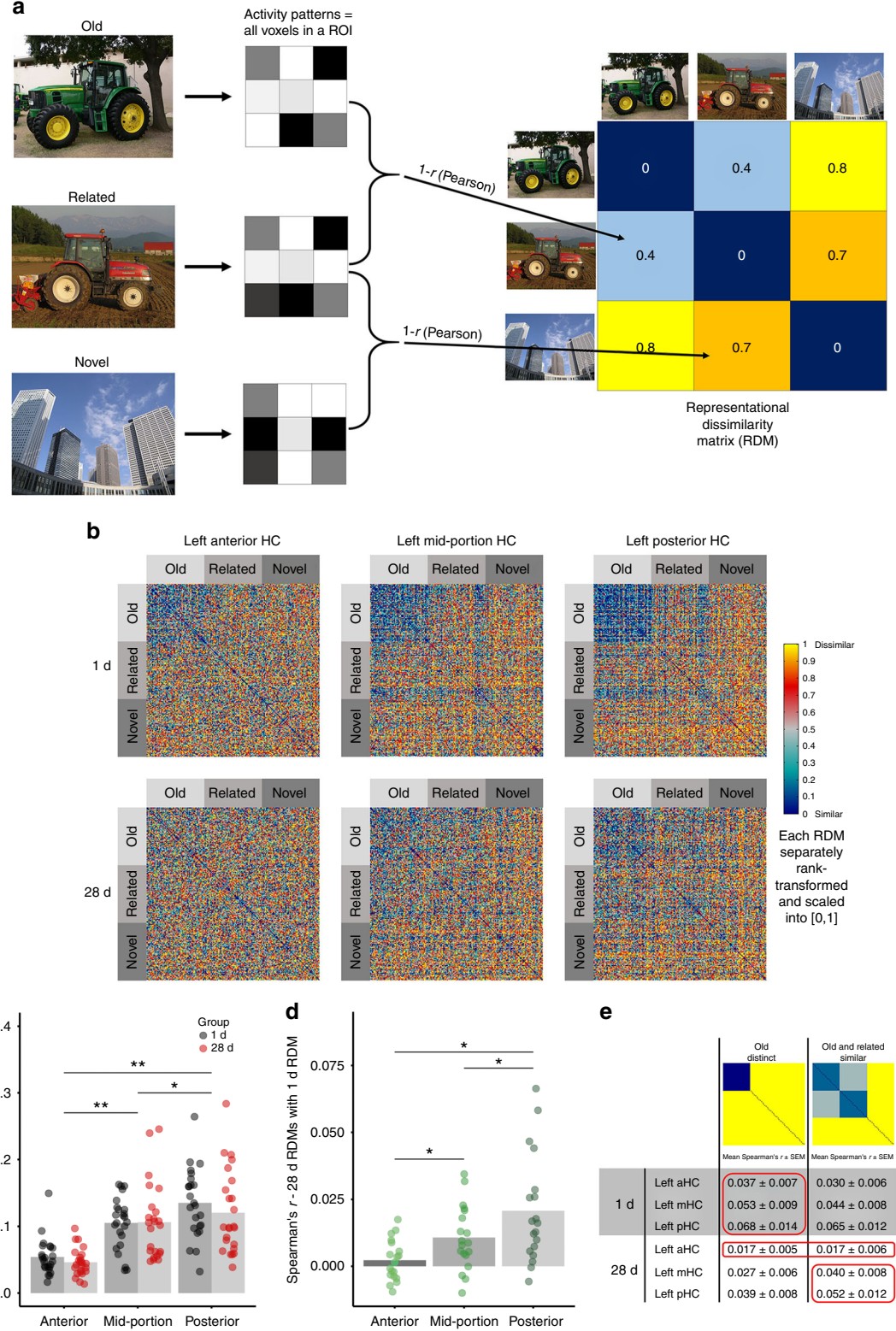

visualizations of these RDMs suggest the most similar activity patterns for combinations of old pictures in all hippocampal subregions in the 1 d group; whereas, in the 28 d group the representational pattern was less clear in the aHC and less-specific in the pHC. Note, however, that for the visualizations each RDM was separately rank transformed and scaled into [0, 1] preventing a direct descriptive comparison across hippocampal subregions. We therefore extracted the mean pattern similarity for each RDM: this showed that the mean similarity was highest in the pHC and lowest in the aHC (main effect HC Long Axis: $F_{(2,92)} = 84.37$, $p = 1.54e-21$, generalized $\eta^2 = 0.350$, $n = 48$; Fig. 4c), suggesting particularly distinct activity patterns in the aHC for different pictures which might allow for highly specific memories, whereas in the pHC the higher neural similarity across different pictures may reflect a larger degree of overlapping representations. We then looked at the similarity of the RDMs across groups and found that the overlap of the representational patterns of the 1 d- and 28 d-groups was significantly higher in the pHC than in the aHC (main effect HC Long Axis: $F_{(2,36)} = 12.72$, $p = 6.65e-05$, generalized $\eta^2 = 0.231$, $n = 19$; Fig. 4d). This suggests that pHC representational patterns changed less over time than aHC representational patterns did.

In order to directly test whether the time-dependent changes of the memory representations in anterior and posterior hippocampal areas were associated with the transformation from detailed to more gist-like memory, we finally compared brain and model-based RDMs. More specifically, we compared the brain RDMs of the hippocampal subregions with two model RDMs: (1) the model "Old Distinct" expects similar activity patterns for all old pictures that are distinct from patterns for related or novel pictures, a representation expected in areas that help detecting the old pictures as old and separate them from related pictures; (2) the model "Old and Related Similar" expects a more similar pattern for the old and related pictures, that is distinct from patterns for the novel pictures, a representation expected in areas that detect the gist as having been encoded but cannot separate between details (Fig. 4e). Based on the behavioral data, we reasoned that the "Old Distinct" model should, in general, fit better in the 1 d group as these participants still had detailed memories, while the "Old and Related Similar" model might fit better in the 28 d group, as for these participants part of the memories had been transformed to gist-like versions. Our analyses showed that, in the 1 d group, the "Old Distinct" model had, compared to the "Old and Related Similar" model, indeed a marginally better fit in the left aHC (one-tailed paired t-test: $t(23) = 1.58$, $p = 0.0635$, $n = 24$) and a better fit in left mHC (one-tailed paired t-test: $t(23) = 1.91$, $p = 0.0345$, $n = 24$), whereas in the left pHC both models were indistinguishable (one-tailed paired t-test: $t(23) = 0.61$, $p = 0.2737$, $n = 24$). In the 28 d group,

on the other hand, the "Old and Related Similar" model had a better fit than the "Old Distinct" model in the left mHC (one-tailed paired t-test: $t(23) = -2.24$, $p = 0.0174$, $n = 24$) and tended to have a better fit in the left pHC (one-tailed paired t-test: $t(23) = -1.53$, $p = 0.0695$, $n = 24$), while in the left aHC both models were indistinguishable (one-tailed paired t-test: $t(23) = -0.07$, $p = 0.4740$, $n = 24$) and the respective model fits were generally rather low.

The brain data were largely comparable in both hemispheres (see Supplementary Figures 7 and 8 for results in the right HC). Group differences in the univariate analysis and the RSA were not modulated by stimulus emotionality (see Supplementary Figs. 6b and 9). We did, however, find time-dependent connectivity changes with hippocampal seed regions that were modulated by stimulus emotionality for some of the ROIs (Supplementary Table 3).

## Discussion

How memories evolve over time is a fundamental issue of the neuroscience of memory. While previous research focused mainly on time-dependent changes in hippocampal and neocortical contributions to memory[3,9,10,12], here we show a time-dependent reorganization along the hippocampal long axis that is related to the transformation from detailed to gist-like memory. More specifically, our data indicate that the aHC is involved in memory specificity and represents actually encoded events distinctly from semantically related information at short retention intervals but shows a marked decrease in activity at longer retention intervals. Activity in the pHC, in turn, was largely unrelated to memory specificity and did not decrease over time, while pHC representational patterns seemed more gist-like at a longer retention interval.

The present data point to a possible involvement of the aHC in the specificity of memory. Previous data in rodents showed that firing fields of ventral hippocampal (corresponding to aHC in humans) place cells are larger than those in the dorsal hippocampus[25], which might translate into more abstract, large-scale aHC memory representations ([26], see also ref. [14]). However, the finding that an animals' exact location can be decoded from ventral hippocampal activity[27] is in line with the role of the aHC in memory specificity that we propose here. In addition, our results fit to a study showing stronger aHC activity for recent memories than for remote memories[20], to a study showing that aHC carries information about memory contexts in immediate and recent (1 day old) memories[28] and to studies showing a consistent implication of the aHC in memory of specific events[29]. Our results further dovetail with reports showing that the aHC specifically is associated with segregating events[30] and with

**Fig. 4** Representational similarity analysis for the left HC long axis ROIs. **a** Schematic overview of the creation of a representational dissimilarity matrix (RDM; for illustration purposes only 3 pictures) modified from ref. [48]. Images courtesy of Andreas Praefcke (green tractor), Akiyoshi's Room (red tractor) and Morio (skyscrapers). **b** The group average RDMs of the left long axis hippocampal ROIs (1 d group in the first row, 28 d group in the second row). Blue colors = most similar, bright colors = most dissimilar; note that for the visualizations each RDM was separately rank transformed and scaled into [0, 1]. **c** Comparison of mean pattern similarities (Pearsons r) across ROIs: the mean similarity in the hippocampal subregions differed significantly. Note that all $n = 48$ participants are included here. **d** Comparison between groups: correlations (Spearman's r) of each single-subject RDM of the 28 d group to the respective average RDM of the 1 d group. The correlations between the two groups differed significantly in the three ROIs. Note that the data in **b** and **d** are from only 40 participants (1 d group = 21, 28 d group = 19) as the remaining eight participants had different sized RDMs and could therefore not be included in the average RDM for the group comparisons. **e** Comparison with two model RDMs: each cell in the table shows the mean of the correlations (Spearman's r + SEM) of the single-subject brain RDMs with the respective model RDM (first three rows = 1 d group, last three rows = 28 d group; $n = 48$). For each ROI the model with the higher correlation was marked by a red frame, in case of very similar correlations both values were marked. In the 1 d group, the "Old Distinct" model showed trends toward a better fit. In the 28 d group the "Old and Related Similar" model had a better fit in the mHC and a trend toward a better fit in the pHC, while in the aHC both models were indistinguishable and the model fits were generally rather low. For analysis of the right HC long-axis ROIs see Supplementary Fig. 8. *$p < 0.05$, **$p < 0.001$

novelty detection[17,31], both of which requires specific memory representations.

Whereas the activity of the aHC was reduced after 28 d, no such decline was observed for the pHC, suggesting that not all parts of the hippocampus decrease in activity over time. The RSA data, however, suggested a time-dependent change of the representational pattern in the pHC. In the 1 d group the pHC representation was already less-specific than the aHC representation, which corroborates the recent idea that there are complimentary learning systems within the hippocampus with one supporting gist-like representations[32,33]. In the 28 d group, the model RSA data even suggested that the representational patterns in the pHC resemble more gist-like patterns. This result suggests a time-dependent decrease in the specificity of the pHC memory representation. This idea is in line with a recent finding[34], showing that neural patterns of overlapping memories were more similar in the pHC after a week of consolidation. In this study, however, part of the memories were actually overlapping (e.g., same scene with different objects), whereas our study extends this finding by using two different pictures with only the semantic gist as overlap, thereby pointing to a memory transformation process.

Thus, there may be two time-dependent processes that contribute to more gist-like memory: a decrease in the aHC supporting memory specificity and an increase in the unspecificity of the mnemonic representation in the pHC, whose activity remains rather stable over time. Whether one process proceeds or follows the other or whether both occur independently remains to be shown.

Our findings suggest a functional specialization in which aHC representations support detailed memories and pHC representations are more gist-like after a longer time delay. Rodent data, however, suggest that the hippocampal long axis is organized along a gradient[15]. Most human studies did not address the mHC and rather little is known about the properties of this subregion. Our finding that the mHC was both with respect to its association with memory specificity and in terms of decreased activity in between the aHC and pHC is in line with the proposed functional gradient along the hippocampal long axis. Yet, how exactly the proposed different functions of the aHC and pHC are bridged is still unclear and remains a challenge for future research.

It is important to note that while we report this time-dependent reorganization within the hippocampus that was linked to memory transformation, we obtained also evidence for the proposed systems consolidation theory[9,12,24]. Hippocampal activity during recognition testing was significantly lower after 28 days than after 1 day, even for items remembered with high confidence. This latter point opposes the transformation hypothesis[2], which would not expect a reduction in hippocampal involvement for high confident, detailed memory. In addition, hippocampus-amygdala connectivity, known to be implicated in vivid memory[23,35], was reduced in the 28 d group compared to the 1 d group. However, this reduction in functional connectivity with the amygdala seemed to be specific to the aHC. This finding is in line with data suggesting that the aHC is connected to, among other regions, the amygdala, whereas the pHC is connected to areas involved in schematic memory such as the precuneus[14,16]. Thus, the aHC and pHC appear to be part of distinct neural networks that are involved in specific vs. gist-like memory and the observed reorganization along the hippocampal long axis is most likely concerted with the postulated large-scale redistribution (i.e., systems consolidation) of memory.

Finally, we would like to point out that the time-dependent changes reported here cannot be interpreted as a mere indication of a reduction in memory strength. In fact, we have designed this study explicitly to be able to differentiate between a general reduction in memory strength and memory transformation processes. In particular, we included related pictures that allowed us to probe memory specificity. If only memory strength was reduced after 28 d, this should be reflected in a comparable increase in the FA rates for related and novel pictures. We observed, however, a much stronger increase of FAs for related pictures than for novel pictures, which is in sharp contrast to the interpretation of a simple reduction in general memory strength but in line with the proposed transformation from detailed to gist-like memory. In addition, our model RSA data can also not be explained by a general reduction in memory strength. This view would imply that the memory for specific details and the gist memory decrease to a similar extent over time so that the relative representation of old and related items remains over time. Our data, however, show that the "Old Distinct" model best characterized activity in the 1 d group, whereas after 28 d the two models were indistinguishable in the aHC and the "Old and Related Similar" model seemed to fit better in the mHC and pHC. Together, these findings indicate that, in addition to the well-known decline in memory strength over time, there is also a change in the nature of memory, from detailed to more gist-like.

Our findings show that while the involvement of the hippocampus as a whole in memory decreases over time, this decrease is not present in all parts of the hippocampus. However, although there was a hippocampal memory representation even long after encoding, the nature (and origin) of the hippocampal contribution to remembering changed significantly with time. To conclude, we suggest here a time-dependent reorganization within the human hippocampus that is linked to a transformation from detailed to gist-like memory and might operate in tandem with the previously suggested large-scale reorganization of memory that occurs in the brain over time[9,12,24].

## Methods

**Participants.** We tested 48 healthy, right-handed, young adults (24 men, 24 women; age: mean = 23.85 years, SD = 3.28 years) without a history of any psychiatric or neurological diseases, without medication intake or drug abuse and without circumstances preventing an MRI scan. All participants gave written informed consent and received monetary compensation for participation. The study protocol was approved by the ethics committee of the German Psychological Society (DGPs). Participants were pseudo-randomly assigned to the 1 d- or 28 d-group (12 women and 12 men per group). All experiments took place in the afternoon or early evening. The sample size corresponds to other studies on the neural underpinnings of memory processes and an a-priori power calculation with G* Power (http://www.gpower.hhu.de/; $f(U) = 0.5$, $\alpha = 0.05$, $1-\beta = 0.90$) for the decisive interaction effect Group × FA Picture Type (see Behavioral data analysis).

**Study design and experimental paradigm.** Testing took place on two experimental days: Day 1, encoding outside of the scanner and Day 2, recognition memory testing in the MRI scanner. Critically, the time interval between encoding and recognition testing was varied between the two experimental groups: for participants in the 1 d group recognition testing took place one day after encoding, while for participants in the 28 d group recognition memory was tested 28 days after encoding. The testing of the two groups was intermixed, so confounds related to changes in, for instance, the technical environment of the scanner over time cannot explain group differences.

**Stimulus material.** We used 180 pictures of natural scenes and objects as stimulus material. About one third of the pictures were taken from the International Affective Picture System (IAPS[36]), while the others were taken from open internet platforms. Half of the pictures contained emotionally negative scenes or objects while the other half contained neutral contents. Participants rated all pictures at the end of the experiment with respect to picture valence (scale from 0 = negative to 100 = positive, with 50 = neutral) and picture arousal (scale from 0 = not arousing to 100 = very arousing). In retrospect, these data confirmed that neutral pictures ($M = 57.38$, SEM = 0.79) were perceived as neutral and negative pictures ($M = 25.99$, SEM = 1.39) as more negative (paired $t$-test: $t_{(47)} = -18.38$, $p < 0.0001$, Cohen's $d = -2.65$, $n = 48$). Furthermore, negative pictures ($M = 47.50$, SEM = 3.01) had higher arousal ratings than neutral ones ($M = 11.15$, SEM = 1.90; paired $t$-test: $t_{(47)} = 13.42$, $p < 0.0001$, Cohen's $d = 1.94$, $n = 48$).

The 180 pictures were divided into three lists (each 30 negative and 30 neutral pictures): List A and List B contained semantically related pictures, i.e., for each

picture in List A there was a matching picture in List B that carried the same gist (e.g., mowing tractor) but different details (e.g., different brand, color, perspective, and background). List C, on the other hand, contained novel pictures that were not semantically related to either List A or List B pictures. Half of the participants learned List A during encoding and List B pictures were used as related lures and List C pictures as novel lures in the recognition test, while the other half of the participants learned List B during encoding and List A pictures were used as related lures and List C pictures as novel lures in the recognition test.

The semantic relatedness of the stimuli was rated by an independent sample ($n = 12$) on a scale from 1 ("not related") to 10 ("highly related"). Corresponding pictures of List A und List B were rated as highly related ($M = 8.25$, SEM = 0.062), in comparison to List A pictures compared to List C pictures ($M = 2.03$, SEM = 0.056, paired $t$-test: $t_{(19,84)} = 12.86$, $p < 0.0001$, Cohen's $d = 4.18$), or List B compared to List C pictures ($M = 1.98$, SEM = 0.055, paired $t$-test: $t_{(11)} = 14.85$, $p < 0.0001$, Cohen's $d = 4.49$).

**Experimental day 1 (memory encoding)**. On the first experimental day, participants performed three encoding runs. In each run, the 60 pictures from the respective list (either A or B) were presented to the participant in random order on a computer screen, using MATLAB (www.mathworks.com) with the Psychophysics Toolbox extensions[37]. Each picture was presented for 2 s followed by a fixation cross of 1 s. Participants were instructed to memorize the pictures. Immediately after each encoding run a free recall task followed: participants verbally listed all the pictures they could remember while the investigator checked off the named pictures on a list and prompted the participant to a more detailed description in case the description of a picture was inconclusive. In total, the encoding session took about 20 min.

**Experimental day 2 (memory testing)**. On the second experimental day, either 1 d or 28 d after encoding, participants first performed another free recall task outside the scanner and then a recognition task while fMRI measurements were taken. In the recognition task, participants saw the 60 old pictures, 60 related pictures, i.e., new pictures carrying the gist of the old pictures, and 60 novel pictures in random order. Each picture was shown for 3.5 s and participants were asked to indicate ("yes" vs. "no") by button press whether they had seen this picture during the encoding session or not. Critically, participants were informed before the task that some of the pictures may be similar to the original ones. Participants were further explicitly instructed to answer "Yes" only if they thought the picture was exactly the same as the one learned on experimental day 1. After participants' response, their choice was marked by a yellow box around the answer. If they answered "Yes" a confidence rating followed: they were asked to indicate on a 4-point scale how confident (not at all confident, slightly confident, quite confident, or very confident) they were that they had seen the picture on experimental day 1. This rating was shown for 2 s, and again their answer was marked by a yellow box. Each trial was followed by a fixation cross with a jittered presentation time of $7 \pm 2$ s.

**Behavioral data analysis**. To assess the performance in the recognition task in general we compared the percentages of hits for old pictures in a mixed-design ANOVA with Group (1 d vs 28 d) as between-subject factor and Emotionality (negative vs neutral) as within-subject factor. In order to assess the specificity of memory, we further analyzed the percentages of FA for related and novel lures in a mixed-design ANOVA with Group (1 d vs 28 d) as between-subject factor and FA Picture Type (related vs novel) and Emotionality (negative vs neutral) as within-subject factors. To additionally include information about the confidence of the participants when answering correctly, we calculated a Confidence Score by weighting each hit by the respective confidence (not at all confident = 0, slightly confident = 1, quite confident = 2 or very confident = 3), resulting in a score of 0–3 for each hit, before summing up overall hits. The maximum Confidence Score is therefore 180 (60 "very confident" hits). This score was again subjected to a mixed-design ANOVA.

We also analyzed the matching picture pairs (old picture and the corresponding related lure carrying the same gist) by assigning each pair to one of three categories: (1) detailed pairs, for which participants could reliably distinguish between old and related pictures and therefore correctly identified the old picture as old and correctly rejected the related picture as new. (2) transformed pairs, for which participants could not rely on detailed memories but still remembered the gist, as reflected by a FA to the related lure (irrespective of the response to the old picture), and (3) forgotten pairs, for which participants may have forgotten the whole picture (both gist and details), as reflected by a miss for the old picture and a correct rejection for the related picture.

Behavioral data analyses were performed with R version 3.3.2 (https://www.r-project.org/). All $p$-values are two-tailed and Welch's $t$-tests were used as default for between group comparisons[38]. The Shapiro–Wilk normality test was applied to the dependent variables: while the Confidence Score ($W = 0.96$, $p = 0.1246$) was normally distributed, this was not the case for the Hits ($W = 0.87$, $p = 0.0001$) and the FA ($W = 0.90$, $p = 0.0009$). Despite this violation of the normality assumption we applied the above described ANOVAs due to the robustness of these tests against the violation of this assumption[39].

**MRI acquisition**. MRI measurements were obtained with a 3T Skyra scanner (Siemens), equipped with a 32-channel head coil. For the functional images, a 3D echoplanar imaging (EPI) sequence (836 volumes) was used with the following parameters: 36 slices, slice thickness = 3 mm, distance factor 20%, repetition time (TR) = 2500 ms, echo time (TE) = 30 ms, voxel size 3.0 mm isotropic. We additionally acquired a high-resolution T1-weighted anatomical image (TR = 2.5 s, TE = 2.12 ms, 256 slices, voxel size = $0.8 \times 0.8 \times 0.9$ mm) and a magnetic (B0) field map to unwarp the functional images.

**Data preprocessing**. The fMRI data were preprocessed using MATLAB and SPM12 (http://www.fil.ion.ucl.ac.uk/spm/). The first four functional images (10 s) were discarded from the rest of the analysis to allow for T1 equilibration. The remaining 832 functional images were first spatially realigned and unwarped using the field maps, then coregistered to the structural image, followed by a normalization to the MNI space. For the univariate analysis, the images were additionally spatially smoothed using an 8 mm full-width half-maximum Gaussian kernel.

**General linear modeling and whole-brain analysis**. For the univariate analysis, the data were analyzed using general linear modeling (GLM) as implemented in SPM12. Six separate regressors for each of the Picture Type × Emotionality combinations were modeled: old negative, old neutral, related negative, related neutral, novel negative, and novel neutral. The onsets of the confidence ratings were additionally included as a regressor of no interest and all regressors were convolved with the canonical hemodynamic response function. Note that we did not include movement regressors in the GLM, as we used the SPM unwarp function in the data preprocessing instead. A high-pass filter of 128 s was used to remove low-frequency drifts and serial correlations in the time series were accounted for using an autoregressive AR(1) model. To look at whole-brain activation differences between the 1 d- and 28 d-groups, we used a two-sample $t$-test design at second-level modeling.

**ROI analysis**. In addition to the whole-brain analysis, we performed regions of interest analyses that focused on brain areas that have previously been implicated in detailed and more semantic or schema-related memory processes[1,3,9,40]. To this end, we used the following anatomical masks from the Harvard-Oxford atlas using a probability threshold of 50%: hippocampus (left and right), anterior parahippocampal gyrus (left and right), posterior parahippocampal gyrus (left and right), precuneus, angular gyrus (left and right), anterior cingulate gyrus, inferior frontal gyrus pars opercularis (left and right), inferior frontal gyrus pars triangularis (left and right), temporal pole (left and right), and the amygdala (left and right). In addition, we used masks created with MARINA (http://www.bion.de/eng/MARINA.php) for the left and right ventromedial prefrontal cortex. The signal within the ROIs was deconvolved for each of the regressors from the GLM (old negative, old neutral, related negative, related neutral, novel negative, and novel neutral) using a finite impulse response function (FIR) on the time course averaged across all voxels of the ROI as implemented within MarsBar[41] for the first seven repetition times (TRs; 15 s). We choose FIR deconvolutions here to capture the shape of the HRF and allow for differences in this hemodynamic response across regions and participants. For statistical comparisons in R, we extracted the peak response from these FIR time courses: as described in refs. [42,43] the peak response was defined as the average signal over time points whose responses (collapsed across all conditions) did not significantly ($p > 0.05$) differ from the numerical peak tested across all participants. In most ROIs this procedure resulted in the peak response being the average across the 5 s and the 7.5 s time points (exceptions: left and right posterior parahippocampal gyrus and right angular gyrus = only the 5 s time point; left angular gyrus = 2.5 s and 5 s time points; precuneus cortex = 7.5 s and 10 s time points; left and right temporal pole = 5 s, 7.5 s and 10 s time points; in the left and right ventromedial prefrontal cortex the numerical peak did not significantly differ from any of the other time points, as no reliable peak response seems to be present in these two ROIs we did not perform further analysis on this data). Then the average signal over the respective time points was calculated for each condition separately (old negative, old neutral, related negative, related neutral, novel negative, and novel neutral) and used in a mixed ANOVA model, with Group (1 d vs 28 d) as between-subject factor and Picture Type (old vs related vs novel) and Emotion (negative vs neutral) as within-subject factors. In case of violation of the sphericity assumption, Greenhouse-Geisser corrections were applied. The $p$-value threshold was adjusted for multiple comparisons by the numbers of ROIs.

**Differentiation along the hippocampal long axis**. In order to look at differences across the long (anterior–posterior) axis of the hippocampus, we used the procedure described by ref. [26] to divide a hippocampal mask into three parts with approximately equal lengths along the long axis, using the WFU pick-atlas[44,45]: pHC from $Y = -40$ to $-30$, mHC from $Y = -29$ to $-19$, and aHC from $Y = -18$ to $-4$. For these new hippocampal ROIs, we then deconvolved the signal for each of the regressors using a finite impulse response function (FIR), and extracted the peak response from these FIR time courses for statistical comparisons in R as described in detail for the other ROIs above. In all of these hippocampal long axis ROIs this procedure resulted in the peak response being defined as the average across the 5 s and the 7.5 s time points. We then run a mixed ANOVA model on

these peak responses, with Group (1 d vs 28 d) as between-subject factor and Picture Type (old vs related vs novel) and Emotion (negative vs neutral) as within-subject factors in each of the ROIs separately. In addition, we performed another analysis to see if the delay manipulation (1 d vs 28 d) had a different effect on these three hippocampal ROIs, using an ANOVA with Group as between-subjects factor and HC Long Axis (aHC, mHC, and pHC) as within-subject factor.

**Comparison of high- and low-confidence hits**. In order to assess whether the time-dependent decrease in hippocampal activity was modulated by confidence, we created a second model that was based on the behavioral responses and confidence ratings of the participants. We modeled five regressors: High-confidence hits (hits with a confidence rating of 3), low-confidence hits (hits with a confidence rating of 0, 1 or 2), related CR, novel CR, and all incorrect answers (misses + related FA + novel FA + no presses). The onsets of the confidence ratings were again included as a regressor of no interest and all other procedures were the same as in the first GLM. We then deconvolved the signal for the high-confidence hits and low-confidence hits in the hippocampal ROIs using a finite impulse response function (FIR) with MarsBar, and extracted the peak response from these FIR time courses for statistical comparisons in R using the same time points as peak as in the analysis above. We then run a mixed ANOVA model on these peak responses, with Group (1 d vs 28 d) as between-subjects factor and Hit Type (high vs low) as within-subject factor in each of the ROIs separately. Group effects were followed up by post hoc Welchs t-tests comparing the groups (1 d vs 28 d) for each Hit Type separately.

**Correlation with behavior**. We extracted the contrast values from the main effects (condition vs baseline) of the first GLM for each Picture Type (old, related, novel) in each ROI again using the MarsBar toolbox, and computed correlations between behavioral memory scores (percentage of related FA; Confidence Score) and the contrast values for the respective picture types (related > baseline, old > baseline) in each of the hippocampal long axis ROIs outside SPM using R for all participants. We then also performed the correlations for each group (1 d vs 28 d) separately. In a next step, we compared the correlations in the hippocampal long axis ROIs with each other (e.g., correlation in aHC with the correlation in pHC), using the cocor package from R (http://comparingcorrelations.org/). As the related FA value (or Confidence Score) was used in all three correlations, we report the results of comparisons of two overlapping correlations based on dependent groups. The reported values are Pearson and Filon's z-scores.

**Functional connectivity analysis**. We used a generalized form of context-dependent psychophysiological interaction (gPPI)[46] to measure task-dependent connectivity with either the whole hippocampus (left and right) or the aHC or pHC (left and right) as seed regions. In contrast to the standard PPI implementation through SPM, the gPPI toolbox allows the inclusion of more than two task conditions in one PPI model and therefore allows a more flexible analysis: we entered the six task regressors from the first GLM model (old negative, old neutral, related negative, related neutral, novel negative, and novel neutral), plus a PPI Interaction term for each of these regressors, plus the time course from the respective seed region and the confidence ratings as regressor of no interest into our first-level PPI model. For second-level modeling, we entered the following contrast files from the first-level PPI analyses (main effects for PPI Old, PPI Related, and PPI Novel, and the differences contrast PPI old negative > PPI old neutral, PPI related negative > PPI related neutral, PPI novel negative > PPI novel neutral) into two-sample t-tests, comparing the 1 d group and 28 d group. We then applied a small volume correction (SVC) for all our other ROIs (see ROI analysis for a list of the ROIs), to find areas which show a significant difference in their connectivity to the seed region between the two groups. Voxels were regarded as significant when falling below a corrected voxel threshold of 0.05 (FWE) adjusted for the small volume. All areas with $k > 10$ significant voxels were reported.

**Representational similarity analysis**. Independent from the univariate analysis, we carried out a RSA[47] in the hippocampal long-axis ROIs using the rsatoolbox[48]. For each ROI and each subject, brain Representational Dissimilarity Matrices (RDMs) were computed based on a single trial univariate GLM estimated on unsmoothed, normalized functional images. The response-amplitude beta estimate maps associated with each trial were converted into t-maps and used to create vectors of activity patterns for each trial, separately for each ROI. These activity patterns were used to calculate the dissimilarity between two trials by correlation distances ($1-r$, Pearson linear correlation). Next, the dissimilarities based on each combination of trials were placed into the respective cells of the 180 × 180 RDMs (Fig. 4a). Due to technical failure, we did not have functional data for all trials in some of the participants. Thus 8 participants had RDMs of slightly different sizes (in the 1 d group three participants had 179 × 179 RDMs; in the 28 d group two participants had 179 × 179 RDMs, two participants had 178 × 178 RDMs and one participant had a 176 × 176 RDM). For visualization of the RDMS we created average RDMs for each group from the single-subject RDMs (Fig. 4b). As this required RDMs of the same size, the visualizations only include the data of the 40 participants with 180 × 180 RDMs.

**Comparison of pattern similarities across ROIs**. We extracted the mean pattern similarity ($r$) from each single-subject RDM in order to get the overall similarity of activity patterns when performing the recognition task in general, irrespective of picture types, per ROI. We then compared these mean pattern similarities across the hippocampal long axis by conducting an ANOVA with the factors HC Long Axis (aHC, mHC, pHC), and Group (1 d vs 28 d), and post hoc Bonferroni corrected paired t-tests to compare each region to each of the other regions. Note that for this analysis all 48 participants were included.

**Comparison of RDMs between groups**. We next compared the RDMs of the two groups in the respective ROIs: for this we extracted the Spearman correlation for each single-subject RDM of the 28 d group with the group average RDM of the 1 d group. This similarity of the RDMs between groups was then again compared between the hippocampal long axis ROIs with an ANOVA with the factor HC Long Axis (aHC, mHC, and pHC), and post hoc Bonferroni corrected paired t-tests to compare each region to each of the other regions. Note that for this analysis only the 40 participants with 180 × 180 RDMs could be included as we aimed to directly compare the RDMs based on single trials, irrespective of picture category. Therefore all RDMs had to be of the same size for this particular analysis.

**Comparison with model RDMs**. We also compared the brain RDMs to two model RDMs (Fig. 4e) that were based on the expected similarities of the different picture types: the model "Old Distinct" expects similar activity patterns for all old pictures that are distinct from patterns for related or novel pictures and the model "Old and Related Similar" expects a similar pattern for the old and related pictures, that is distinct from patterns for the novel pictures. We calculated Spearman's rank correlation coefficient for each single-subject brain RDM and these a-priori model RDMs. This rank coefficient is beneficial if it is not possible to assume a direct linear match between the RDMs that are compared[47], as is the case here. We then calculated the mean of these Spearman's r's for each group separately (1 d vs. 28 d) to find the model with the overall best fit in each group and ROI. We then statistically compared the two models in each ROI per group with one-tailed paired t-tests, expecting a better fit for the "Old Distinct" model in the 1 d group and, on the other hand, a better fit for the "Old and Related Similar" model in the 28 d group. Note that for this analysis all 48 participants could be included by creating model RDMs of the respective size matching each participants brain RDM.

**Data availability**. All data and codes are available from the corresponding authors upon request. The data are not publicly available yet because they contain information that could compromise research participant privacy and consent. In the near future, they will be de-identified at the level of contemporary best practices and made publicly available, together with relevant code, at the corresponding author's GitHub repository (https://github.com/LarsSchwabeHamburg/transformation).

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

## Acknowledgements

The authors thank Daniel Kutzner, Wiebke Schmidt, Sandra Erbach, Kristin Medel, and Amina Shah for assistance during data collection and Dr. Lynn Nadel as well as Dr. Tobias Sommer for helpful comments on a previous version of this manuscript. This work was supported by the German Research Foundation (DFG; SCHW1357/12-1).

## Author contributions

L.C.D. collected the data, analyzed the data, and wrote the manuscript. L.S. conceived and designed the experiment, supervised the project, and wrote the manuscript.

## Additional information

**Competing interests:** The authors declare no competing interests.

