## [Peer Review File(PDF 647 kb) · Nature Communications]

Reviewers' comments:

Reviewer #1 (Remarks to the Author):

There is long-standing evidence of a functional reorganization of memory between the hippocampus and neocortical structures over time. Recently also complementary learning systems within the hippocampus have been proposed. Dandolo & Schwabe report a time-dependent reorganization of memories within the hippocampus, demonstrating detailed and unique representations in the anterior hippocampus (aHC) that declined over time, and more gist-like representations in the posterior hippocampus (pHC), particularly after a longer retention interval. Their study demonstrates different learning systems within the hippocampus and thus greatly contributes to our understanding of how memory is stored and transformed after encoding.

The experiment is well designed and the manuscript is well written. The strong behavioral results that are well supported by findings in brain activity make this a particularly interesting work. I have only a number of minor points that may clear up some remaining questions.

1. Whole brain analysis:

- Were interaction analyses timepoint x picture type performed? Please report.
- ll. 221ff: If interaction analyses not significant, please add note of caution when stating that only activity in the aHC declined over time.
- Supplementary table 1:

A glass brain view would be helpful in addition to the contrast tables.

Precuneus is listed as Precuneous Cortex. Please keep consistent with manuscript.

Can closest regions be listed instead of/in addition to "Location not in atlas"?

2. PPI analysis: ll. 129ff, 146ff, Fig. 3, Suppl. Fig. 3

- It did not become completely clear to me how the finding of reduced connectivity between hippocampus and amygdala relates to the remaining results. Was this an a priori hypothesis?
- Since activity in the amygdala is traditionally associated with emotional processing: did connectivity between these regions depend on emotionality of the pictures?
- ll. 146ff, Fig. 3: "Specifically the right aHC-right amygdala connectivity for related pictures..." was an interaction analysis timepoint x picture type performed? Please report.

3. Correlation with behavior:

ll. 153f, Fig. 2: Were correlations between FA Related and Contrast Values and correlations between Confidence Score and Contrast Values compared statistically between ROIs? This is crucial to support the statement that the aHC is linked to memory specificity, whereas pHC is not. Please report and specify how comparison was conducted.

4. RSA analysis:

How was similarity/Spearman correlation compared between subregions and groups (Fig. 4 c and d)? Please specify in Methods section (also see below).

5. RSA model comparisons:

How was "model fit" compared (Fig. 4 e)? If no statistical test were conducted, a note of caution is warranted in both the Results as well as the Discussion section.

6. Methods:

The Methods section could benefit from detail:

- ll. 370 ff: Why where no movement regressors included in the first level design matrix?
- ll. 380: Rationale behind choosing FIR deconvolution over parameter estimates to compare activity between conditions in ROIs should be briefly explained. Was the timecourse averaged over all voxels in ROI?
- ll. 399ff: why were parameter estimates chosen to compare responses for high confidence vs. low confidence hits? Correction for multiple comparisons? Rationale behind using Welch tests?
- ll. 453ff Please specify how similarity was compared statistically (also see above)
- ll. 458ff Please specify how RDMs were compared statistically (also see above)
- ll. 463ff Please specify how exactly "model fit" was calculated? Simple Spearman correlation? Why is this a good measure? How was model fit compared statistically? (also see above)

7. Supplement:

Suppl. Figs. 5 and 7 nicely replicate the findings for left hippocampus reported in the main paper in the right hippocampus. Why not mention this explicitly also in the Suppl. Figure legends?

Reviewer #2 (Remarks to the Author):

The study investigated a timely and interesting question and the procedure was clearly described and well-suited to address the authors' questions. The main behavioral finding that after 28 days, there are greater false alarms to related lures is very interesting and is a clever means to tease apart detailed versus gist-level memories. However, in many behavioral and neural analyses, the authors did not provide sufficient statistical tests to support their conclusions, and often over-interpreted findings. I have provided a few examples below:

Figure S2 shows main effects of emotion, where negative images are remembered better and with higher confidence than neutral at both time points. For related memories, they find that the change in proportion of false alarms from 1d to 28d is even greater for negative images than for neutral images (S2C) and that there are higher proportions of detailed and transformed memories, and a lower proportion of forgotten memories, for negative relative to neutral images. This is an interesting finding, but it does not support the authors' interpretation in the main text that 'stimulus-related emotional arousal strengthens later retention mainly through an enhanced gist memory representation'. The data demonstrate the majority of items are remembered and their related lures are rejected (which is enhanced for negative memories), while there seems to be a marginal improvement in retention due to more transformed negative memories. Further, the critical pairwise t-tests

of whether there are actually more transformed memories for negative vs neutral memories at each time point are not reported.

Tables S1: These whole-brain contrasts reveal lower activity in a broad swath of regions for 28d versus 1d, across all three conditions. However, no claims can be made about the specificity of the decrease by picture type, unless contrasts of Remembered > Transformed, etc are reported separately for each group. Further, no reported clusters survive multiple comparisons. Table S2: the group x time bin effect is not influenced by condition, which suggests that activity in the hippocampus and amygdala is lower on day 28 regardless of picture type. Figures 3, S3 and S4 also suggest very little effect of condition on the decreased activity and connectivity in the 28d group, and there are no ANOVAs reported to provide evidence for an interaction between behavior, group and neural measure. The authors state that this is 'generally in line with the systems consolidation view, suggesting that hippocampal involvement in memory decreases over time', but none of the findings suggest a decrease that is specific to remembered or transformed items. Rather, there seems to be some other task- or group-level difference that is influencing all conditions. Figure S5A, Table S2, and others: ANOVAs are not statistically valid means to test differences in FIR timecourses, as the time points are not independent of each other due to the inherent temporal correlations in BOLD signal. A more appropriate analysis would be to conduct statistics over the peak of the HRF (Schmitz et al, JNeuro, 2010; Turk-Browne et al., JNeuro, 2012) or conduct a permutation test.

Figure 2C, 2D, S5B, and others: It is not valid to collapse across 1d and 28d groups in these scatterplots. The correlations are likely driven by the main effect of day on % of related false alarms and confidence, and the corresponding main effect of day on activity in anterior and mid hippocampus. Computing these correlations separately for 1d and 28d is the appropriate test of a correlation between BOLD activity and behavior.

If these statistical issues are appropriately addressed, the finding that anterior hippocampus exhibits decreased activity at 28d relative to 1d shows promise. However, stronger evidence for a time-dependent change in memory specificity would be to examine whether hippocampal involvement at 28d decreases specifically for transformed items, and that such a decrease is not present for remembered items (or high-confident remembered items), or for forgotten items. As currently reported, there is no evidence that these effects are not a task- or group-level difference.

Figure 4E: There is no formal statistical test between model fits for the old/distinct versus old/related similar RDM. Without such a test, the interpretation that the anterior hippocampus best fits the old/distinct model and the posterior hippocampus best fits the old/related similar model is unfounded.

Other comments:

It is unclear why left HC activation across the long axis is reported in Figure 2, but right HC activation is reported in Figure S5 even though the results are close to identical.

The reason for excluding 24 trials due to lack of data in a subset of participants is unclear. All data could be included with some changes to the analysis pipeline. RDMs could be generated using a weighted average to account for the number of trials per participant. Or,

permutation tests could be developed to account for the different bin sizes across participants.

A discussion of similar papers showing time-dependent transformations of memories should be included. Both papers report distinctions along the hippocampal long axis.

Ritchey, Maureen, Maria E. Montchal, Andrew P. Yonelinas, and Charan Ranganath. "Delay-Dependent Contributions of Medial Temporal Lobe Regions to Episodic Memory Retrieval." *eLife* 4 (January 13, 2015): e05025. <https://doi.org/10.7554/eLife.05025>.

Tompary, Alexa, and Lila Davachi. "Consolidation Promotes the Emergence of Representational Overlap in the Hippocampus and Medial Prefrontal Cortex." *Neuron* 96, no. 1 (September 27, 2017): 228–241.e5. <https://doi.org/10.1016/j.neuron.2017.09.005>.

Reviewer #3 (Remarks to the Author):

The current report investigates brain responses, especially hippocampal responses, during a recognition memory test at two different delay intervals following encoding: 1 day and 28 days. The authors report novel and interesting interactions along the long hippocampal axis for responses at these intervals. I did feel the authors overstated the implications of their findings. I think it would be appropriate to be more cautious about both what was observed and the nature of the variables involved. While I thought the methods were mostly technically sound, I thought greater transparency was needed in some of the measures.

Main comments

- The authors conclude that "The present data point to a critical involvement of the aHC" in gist. Given that the data are correlational, I'm not sure how they would do so. At best they can indicate a "possible involvement".
- The authors use a temporal manipulation. When were the 1-day and 28-day scans gathered relative to one another? Were scans of each type intermixed to ensure there was no confound associated with changes to the scanner technical environment over time?
- The authors transformed each RDM using rank transform and scaling. The authors point out that this prevents comparison across regions, which is correct - these transformations also make it impossible to ascertain the degree of similarity observed at each time in each region, or even whether similarity values are just random distributions around zero. Higher rank values could also be expected for values that correspond to noise if similarities are in general negative, which is not unheard of. It's not clear why the authors would use these transforms - I suggest they use the pattern similarity r-values that they use in their computations for sake of transparency.

- At various points the authors describe these rank-transformed values as e.g., "pHC memory representations". Given how indirect the measures are, this is quite optimistic. It would be far more appropriate to describe the DVs as what they actually are, e.g., "pattern similarity values drawn from pHC", as opposed to the construct the authors are hoping they measure.
- The authors argue that findings of more general memories, weakening hippocampal responses, and lower specificity of pattern similarity in pHC, are a clear indication of memory being transformed into gist representations. However, they do not explore reasonable theoretical alternatives. Can't this be explained more simply as a reduction in memory strength? This possibility should at least be discussed.
- Likewise, alternatives to systems consolidation should be considered. The lack of reduction in cortical activity is not surprising given that it is a recognition test and stimuli are being directly shown to participants. The reduction in hippocampus activity could reflect improvement in memory, since the delayed stimulus presentation is essentially a repetition of the encoding phase, and better memory could correspond to more repetition suppression.
- Why were the behavioural data not tested for normality even though tests that require normality were employed?
- In figures, activity is described as a contrast value, but it is not clear what is being contrasted against in each case. This is important for interpretation and should be made clear in both the text and graphics.

Minor comments

- I don't believe "(25, see also13)" is the right citation formatting.
- Fig. 4b is too small to be useful or see properly.
- "consistent implication of the aHC in the recall of specific events^{19, 27}." But the current study investigated recognition, not recall.

Responses to reviewers

Reviewer 1:

We thank the reviewer for his/her positive and constructive comments. We are glad that he/she considers our experiment well designed, the manuscript well written and our findings interesting.

1 Whole brain analysis:

1a) Were interaction analyses timepoint x picture type performed? Please report.

RESPONSE: Yes, timepoint x picture type interactions were performed, and were significant in the following ROIs: left and right hippocampus, left and right amygdala, left and right inferior frontal gyrus pars opercularis, left and right angular gyrus, precuneous cortex, right temporal pole and anterior cingulate gyrus. However, in response to a comment from Reviewer 2 we have now conducted our univariate ROI analysis on the peak responses of the time course and not the whole FIR time course. Therefore, the factor time point is not included in the now reported analyses anymore. We have, however, now included the Picture Type main effect in our Supplementary Table 2. This main effect is also significant in the most relevant of the above listed ROIs (left and right hippocampus, left and right amygdala, left and right inferior frontal gyrus pars opercularis and anterior cingulate gyrus).

1b) II. 221ff: If interaction analyses not significant, please add note of caution when stating that only activity in the aHC declined over time.

RESPONSE: We were not entirely sure whether the reviewer was referring to the interactions for the whole-brain ROI analyses or to the analyses of activity along the hippocampal long-axis. For the whole-brain analyses, we analyzed now the peak responses (please see our response to comment 1a). For the hippocampal long axis, we had not directly compared the different group effects within the three hippocampal long axis ROI statistically. In response to this comment, however, we now performed an ANOVA with Group as between-subjects factor and Hippocampal Long Axis ROI as within-subject factor with the peak response of the FIR time course as dependent variable. The Group x HC Long Axis interaction was significant (both in the left and right HC). The “post-hoc tests” to this interaction are the group differences (1d vs 28d) in each ROI separately (significant in the aHC and mHC, but not in the pHC), that we had already reported and referred to in the above stated argument.

Please see the methods section for a description of the newly added ANOVA, page 15, lines 468 to 471:

“In addition, we performed another analysis to see if the delay manipulation (1d vs 28d) had a different effect on these three hippocampal ROIs, using an ANOVA with Group as between-subjects factor and HC Long Axis (aHC, mHC and pHC) as within-subject factor.”

Please see the results section for the result of this ANOVA, page 5, lines 138 to 139:

“These differences between the ROIs were underlined by a significant Group x HC long axis interaction ($F(2,92) = 6.07$; $p = 0.0033$, generalized $\eta^2 = 0.036$).”

1c). *Supplementary table 1: A glass brain view would be helpful in addition to the contrast tables. Precuneus is listed as Precuneous Cortex. Please keep consistent with manuscript. Can closest regions be listed instead of/in addition to "Location not in atlas"?*

RESPONSE: We agree and have now added glass brains displaying the activations indicated in Supplementary Table 1 in a new Supplementary Figure 3. Furthermore, we now use the label "Precuneous Cortex" consistently throughout the manuscript. Finally, we used the AAL2 atlas to label regions that were previously listed as "Location not in atlas" based on the Harvard Oxford Atlas (please see the changes in Supplementary Table 1 and in the legend of Supplementary Table 1).

2. *PPI analysis: ll. 129ff, 146ff, Fig. 3, Suppl. Fig. 3*

2a) *It did not become completely clear to me how the finding of reduced connectivity between hippocampus and amygdala relates to the remaining results. Was this an a priori hypothesis?*

RESPONSE: We used the hippocampus (and later the aHC or pHC) as seed region and looked at all of our other ROIs (see Table S2) in the gPPI analysis. We found significant differences in connectivity between the 1d and 28d-group only with the amygdala. Although not a-priori predicted, this finding is of particular interest because amygdala-hippocampus connectivity is known to play a crucial role for vivid memories. We make the relation of this connectivity finding to the remaining results more explicit now. Please see page 5, lines 115 to 121:

"In addition, we performed a psychophysiological interaction (PPI) analysis to test whether the crosstalk of the hippocampus with other areas critical for memory formation changed as a function of time. Our analysis showed specifically reduced functional connectivity between the right hippocampus and the right amygdala in the 28d-group compared to the 1d-group (Supplementary Fig. 4). This decrease in hippocampal-amygdala connectivity was of particular interest as the interaction of these areas is commonly linked to vivid memory²²."

2b) *Since activity in the amygdala is traditionally associated with emotional processing: did connectivity between these regions depend on emotionality of the pictures?*

RESPONSE: We initially did not test for an influence of the emotionality of the pictures in this analysis because we did not find Group \times Emotion interactions in the univariate fMRI results, and therefore collapsed the data across emotions in the following analyses. However, in response to this comment we have now added the first level difference contrasts *PPI Old Negative > PPI Old Neutral*, *PPI Related Negative > PPI Related Neutral*, *PPI Novel Negative > PPI Novel Neutral* to our gPPI analysis. The results of all gPPI analyses are now reported in Supplementary Table 3. The difference in connectivity between the groups for the right HC (right aHC) with the right amygdala that we found for the main effects (Shown in Figure 3 and Supplementary Figure 4) were not influenced by the emotionality. Instead, other areas showed significant differences between the two groups in connectivity to the hippocampal seed regions when using the Negative > Neutral contrasts (e.g. left and right inferior frontal gyrus pars opercularis, left and right temporal pole, anterior cingulate gyrus). For the amygdala there was an influence only on the **left** HC – right amygdala connectivity (in comparison to the **right** HC – right amygdala connections we found for the main effects). We refer the reader to these additional results on page 8 lines 220 to 223:

"Group differences in the univariate analysis and the RSA were not modulated by stimulus emotionality (see Supplementary Fig. 6 and 8). We did, however, find time-dependent connectivity changes with hippocampal seed regions that were modulated by stimulus emotionality for some of the ROIs (see Supplementary Table 3)."

Please see page 16 lines 507 to 512 for changes in the methods section:

“For 2nd-level modelling, we entered the following contrast files from the 1st-level PPI analyses (main effects for *PPI Old*, *PPI Related* and *PPI Novel*, and the differences contrast *PPI Old Negative > PPI Old Neutral*, *PPI Related Negative > PPI Related Neutral*, *PPI Novel Negative > PPI Novel Neutral*) into two-sample t-tests, comparing the 1d-group and 28d-group. We then applied a small volume correction (SVC) for all our other ROIs (see ROI analysis for a list of the ROIs), to find areas which show a significant difference in their connectivity to the seed region between the two groups.”

2c) ll. 146ff, Fig. 3: “Specifically the right aHC-right amygdala connectivity for related pictures...” was an interaction analysis *timepoint x picture type* performed? Please report.

RESPONSE: In these analysis we did not perform FIR deconvolutions, but used the standard gPPI analysis in combination with Small Volume Correction. The results were significant for *related* pictures but not *old* or *novel* pictures in the analysis with the right aHC as seed. We now made this clearer in the results section, see page 5, lines 140 to 143:

“Moreover, connectivity analysis using the aHC and pHC as seed regions revealed that it was specifically the right aHC-right amygdala connectivity for *related* pictures (but not *old* or *novel* pictures) that was reduced in the 28d- relative to the 1d-group (SVC peak level: $x = 20$, $y = -4$, $z = -14$, $t = 3.59$; $p(\text{FWE}) = 0.0156$, $k=15$; Fig. 3).”

3. Correlation with behavior:

ll. 153f, Fig. 2: Were correlations between FA Related and Contrast Values and correlations between Confidence Score and Contrast Values compared statistically between ROIs? This is crucial to support the statement that the aHC is linked to memory specificity, whereas pHC is not. Please report and specify how comparison was conducted.

RESPONSE: We agree and compared the correlations now explicitly between ROIs. These comparisons show that the correlation between the Percentage of FA *related* and the respective values in the ROIs differed significantly between the aHC and pHC (left: $z = -3.46$, $p = 0.0005$; right: $z = -2.18$, $p = 0.0291$). As one would expect given the “in-between position” of the mHC, the correlation differences were less clear cut for the mid-portion: aHC and mHC (left: $z = -2.12$, $p = 0.034$, right: $z = -2.48$, $p = 0.0133$), mHC and pHC (left: $z = -2.00$, $p = 0.0460$; right: $z = -0.98$, $p = 0.3295$). The correlations between the Confidence Score and the respective values in the ROIs also differed significantly between the aHC and pHC (left: $z = 2.81$, $p = 0.0049$; right: $z = 2.28$, $p = 0.0266$), with again mixed results when looking at the comparison between mHC and the other two ROIs: aHC and mHC (left: $z = 1.36$, $p = 0.1736$; right: $z = 2.00$, $p = 0.0458$), mHC and pHC (left: $z = 2.40$, $p = 0.0167$; right: $z = 1.56$, $p = 0.1194$). Thus, the important distinction between anterior and posterior HC is clearly significant. This important result is reported on page 6, lines 152 to 154:

“... and the correlations between activity and indicators of memory specificity were significantly distinct in the left aHC and pHC (FA *related*: $z = -3.46$, $p = 0.0005$; Confidence Score: $z = 2.81$, $p = 0.0049$).”

And in the Supplementary Figure 7 Legend for the right HC:

“The correlations across all participants were significantly distinct in the right aHC and pHC ($z = -2.18$, $p = 0.0291$).”

The method on how these comparisons were conducted is described in the methods part, page 16, lines 492 to 497:

“In a next step, we compared the correlations in the hippocampal long axis ROIs with each other (e.g. correlation in aHC with the correlation in pHC), using the *cocor* package from R (<http://comparingcorrelations.org/>). As the *related* FA value (or Confidence Score) was used in all three correlations, we report the results of comparisons of two overlapping correlations based on dependent groups. The reported values are Pearson and Filon's z-scores.”

4. RSA analysis:

How was similarity/Spearman correlation compared between subregions and groups (Fig. 4 c and d)? Please specify in Methods section (also see below).

RESPONSE: To compare the mean pattern similarities across ROIs, we conducted an ANOVA with the factors hippocampal long axis (anterior, mid-portion, posterior), and group (1d vs 28d), and post-hoc Bonferroni corrected paired t-tests to compare each region to each of the other regions (Fig 4c). And to compare the similarity of the RDMs between groups across the ROIs, we used an ANOVA with the factor hippocampal long axis (anterior, mid-portion, posterior), and post-hoc Bonferroni corrected paired t-tests to compare each region to each of the other regions (Fig 4d). We have now described this more clearly in the methods sections.

Please see on page 17, lines 536 to 539:

“We then compared these mean pattern similarities across the hippocampal long axis by conducting an ANOVA with the factors HC Long Axis (aHC, mHC, pHC), and Group (1d vs 28d), and post-hoc Bonferroni corrected paired t-tests to compare each region to each of the other regions.”

And on page 17, lines 543 to 546:

“This similarity of the RDMs between groups was then again compared between the hippocampal long axis ROIs with an ANOVA with the factor HC Long Axis (aHC, mHC, pHC), and post-hoc Bonferroni corrected paired t-tests to compare each region to each of the other regions.”

5. RSA model comparisons:

How was “model fit” compared (Fig. 4 e)? If no statistical test were conducted, a note of caution is warranted in both the Results as well as the Discussion section.

RESPONSE: We compared the model fits now directly with directed paired t-tests. The results are reported on pages 7, lines 208 to 218:

“Based on the behavioral data, we reasoned that the “*Old Distinct*” model should, in general, fit better in the 1d-group as these participants still had detailed memories, while the “*Old and Related Similar*” model might fit better in the 28d-group, as for these participants part of the memories had been transformed to gist-like versions. Our analyses showed that, in the 1d-group, the “*Old Distinct*” model had, compared to the “*Old and Related Similar*” model, indeed the better fit in the left aHC ($t(23) = 1.86, p = .037$, one-tailed) and left mHC ($t(23) = 1.96, p = .031$, one-tailed), whereas in the left pHC both models were indistinguishable ($t(23) = -0.11, p = .542$, one-tailed). In the 28d-group, on the other hand, the “*Old and Related Similar*” model had a better fit than the “*Old Distinct*” model in the left mHC ($t(23) = -2.61, p = .008$, one-tailed) and a trend towards a better fit in the left pHC ($t(23) = -1.42, p = .085$, one-tailed), while in the left aHC both models were indistinguishable ($t(23) = -0.32, p = .38$, one-tailed) and the model fits were generally rather low.”

Moreover, we went over the discussion section again to tone down our interpretations of the model RSA data, where appropriate. Please see, for example, page 8, lines 251 to 252:

“...the model RSA data suggested that the representational patterns in the pHc resemble more gist-like patterns at the 28d retention interval.”

6. *Methods:*

6a) *ll. 370 ff: Why were no movement regressors included in the first level design matrix?*

RESPONSE: As we had fieldmaps available and a few subjects moved more than 1mm/1°, we used the SPM “Realign & Unwarp” function to deal with movement. When one uses unwarping it is disadvantageous to include the movement regressors in the first level design matrix, as this would “overwrite” the unwarping process and would have therefore made the unwarping unnecessary, at the same time sacrificing the benefits of the unwarping. We now added a short note in the methods part to make the reason for not adding the movement regressors clearer to the reader. Please see page 14, lines 421 to 422:

“Note that we did not include movement regressors in the GLM, as we used the SPM unwarp function in the data preprocessing instead.”

6b) *ll. 380: Rationale behind choosing FIR deconvolution over parameter estimates to compare activity between conditions in ROIs should be briefly explained. Was the timecourse averaged over all voxels in ROI?*

RESPONSE: We choose FIR deconvolutions to capture the shape of the HRF and allow testing for differences in this hemodynamic response across regions and participants. The time course was averaged over all voxels in a ROI. We now explicitly stated this in the methods section. Please see page 14, lines 439 to 441:

“We choose FIR deconvolutions here to capture the shape of the HRF and allow for differences in this hemodynamic response across regions and participants.”

6c) *ll. 399ff: why were parameter estimates chosen to compare responses for high confidence vs. low confidence hits? Correction for multiple comparisons? Rationale behind using Welch tests?*

RESPONSE: We agree, that our use of methods so far was inconsistent (FIR analysis for *old vs related* vs *novel* and a standard GLM analysis reporting parameter estimates for High Confidence Hits vs Low Confidence Hits) and therefore potentially confusing for the reader. We therefore decided now to use the FIR peak response analysis for all the univariate ROI analysis, in order to be consistent across analyses. We have therefore changed the analysis of the High Confidence Hits vs Low Confidence Hits accordingly. Please see Supplementary Figure 5. The main results and conclusions of this analysis have not changed from the previously reported parameter estimates (The decrease in activity in the 28d-group in comparison to the 1d-group is significant for High Confidence Hits in the overall hippocampal ROI, and this decrease is most pronounced in the aHC). We also report the correction for multiple comparisons for the hippocampal long axis ROIs. (Please see the Figure Legend of Supplementary Figure 5). We now use ANOVAs and post-hoc Welch's t-tests to analyse the data extracted from the FIR time courses, as we do for the other univariate ROI-analysis. Note that Welch's t-tests were used as default for all between group comparisons, as has been recently recommended (Delacre, M., Lakens, D. & Leys, C., (2017). *Why Psychologists Should by Default Use Welch's t-test Instead of Student's t-test. International Review of Social Psychology*) as they do not depend on the assumption of homogeneity of variance. Please see pages 15 and 16, lines 479 to 485 for a description of the now used methods:

“We then deconvolved the signal for the High Confidence Hits and Low Confidence Hits in the hippocampal ROIs using a finite impulse response function (FIR) with MarsBar, and extracted

the peak response from these FIR time courses for statistical comparisons in R using the same time-points as peak as in the analysis above. We then run a mixed ANOVA model on these peak responses, with Group (1d vs 28d) as between-subjects factor and Hit Type (high vs low) as within-subject factor in each of the ROIs separately. Group effects were followed up by post-hoc Welch's t-tests comparing the groups (1d vs 28d) for each Hit Type separately."

6d) II. 453ff Please specify how similarity was compared statistically (also see above)

RESPONSE: To compare the mean pattern similarities across ROIs we conducted an ANOVA with the factors hippocampal long axis (anterior, mid-portion, posterior), and group (1d vs 28d), and post-hoc Bonferroni corrected paired t-tests to compare each region to each of the other regions. We have now described this in the methods sections. Please see on page 17, lines 536 to 539:

"We then compared these mean pattern similarities across the hippocampal long axis by conducting an ANOVA with the factors HC Long Axis (aHC, mHC, pHC), and Group (1d vs 28d), and post-hoc Bonferroni corrected paired t-tests to compare each region to each of the other regions."

6e) II. 458ff Please specify how RDMs were compared statistically (also see above)

RESPONSE: To compare the similarity of the RDMs between groups across the ROIs we used an ANOVA with the factor hippocampal long axis (anterior, mid-portion, posterior), and post-hoc Bonferroni corrected paired t-tests to compare each region to each of the other regions. We have now described this more clearly in the methods sections. Please see on page 17, lines 543 to 546:

"This similarity of the RDMs between groups was then again compared between the hippocampal long axis ROIs with an ANOVA with the factor HC Long Axis (aHC, mHC, pHC), and post-hoc Bonferroni corrected paired t-tests to compare each region to each of the other regions."

6f) II. 463ff Please specify how exactly "model fit" was calculated? Simple Spearman correlation? Why is this a good measure? How was model fit compared statistically? (also see above)

RESPONSE: We used the Spearman's rank correlation coefficient to calculate the fit of each model to the brain RDMs, as suggested for comparisons with model RDMs by Kriegeskorte et al. 2008 and used in other memory RSA studies (e.g. Backus et al. 2015). This measure is beneficial when a linear match between dissimilarity matrices cannot be assumed. We have now added this information to the methods section. Please see on page 18, lines 552 to 555:

"We calculated Spearman's rank correlation coefficient for each single subject brain RDM and these a-priori model RDMs. This rank coefficient is beneficial if it is not possible to assume a direct linear match between the RDMs that are compared⁴⁵, as is the case here."

We now compared the model fits directly with directed paired t-tests. Please see our response to your comment 5.

7. Supplement:

Suppl. Figs. 5 and 7 nicely replicate the findings for left hippocampus reported in the main paper in the right hippocampus. Why not mention this explicitly also in the Suppl. Figure legends?

RESPONSE: Thanks for this suggestion. We have changed the Supplementary Figure legends accordingly.

Reviewer 2:

We thank the reviewer for his/her constructive and helpful comments. We are glad that he/she considers the question we addressed timely and interesting.

1. Figure S2 shows main effects of emotion, where negative images are remembered better and with higher confidence than neutral at both time points. For related memories, they find that the change in proportion of false alarms from 1d to 28d is even greater for negative images than for neutral images (S2C) and that there are higher proportions of detailed and transformed memories, and a lower proportion of forgotten memories, for negative relative to neutral images. This is an interesting finding, but it does not support the authors' interpretation in the main text that 'stimulus-related emotional arousal strengthens later retention mainly through an enhanced gist memory representation'. The data demonstrate the majority of items are remembered and their related lures are rejected (which is enhanced for negative memories), while there seems to be a marginal improvement in retention due to more transformed negative memories. Further, the critical pairwise t-tests of whether there are actually more transformed memories for negative vs neutral memories at each time point are not reported.

RESPONSE: We agree that our wording describing the emotion effects were potentially misleading. We have therefore changed the description in the text accordingly. Please see on page 4, lines 102 to 108:

“Our behavioral data further suggest that stimulus-related emotional arousal influenced the transformation to gist-like memories: after 28 days significantly fewer negative pictures were forgotten than neutral ones ($t(23) = -5.00$, $p < 0.0001$, Cohen's $d = -1.02$) and, even more interestingly, negative pictures were significantly more often transformed than neutral ones ($t(23) = 2.67$, $p = 0.0138$, Cohen's $d = 0.54$; Supplementary Fig. 2), in line with findings^{20,21} suggesting that superior memory for emotional material, indicated here by the slower forgetting rate, comes at the cost of reduced memory for contextual details, reflected here in an increase in transformed memories.”

We have also rearranged the Supplementary Figure 2, and the legend to this figure, to show the critical pairwise t-tests comparing negative to neutral pictures in each group separately. We hope that the figure is clearer now and it is now easier to understand the emotionality effects.

2. Tables S1: These whole-brain contrasts reveal lower activity in a broad swath of regions for 28d versus 1d, across all three conditions. However, no claims can be made about the specificity of the decrease by picture type, unless contrasts of Remembered > Transformed, etc are reported separately for each group. Further, no reported clusters survive multiple comparisons.

RESPONSE: We completely agree that we cannot make any claims about the specificity of the decrease by picture type. However, we did not aim to make this claim in this univariate whole-brain analysis. In all three trial types, participants are required to engage their memory to make a correct response. Here we just wanted to show the difference in brain activity between the groups when they perform the memory task, irrespective of picture type (please see also our next response for a further discussion of this point). It is further correct that no reported clusters survive the correction for multiple comparisons, as we stated explicitly in the Figure legend. Nevertheless, we report these results for future reference and potential usage of our data, for instance, in meta-analyses.

3. Table S2: the group x time bin effect is not influenced by condition, which suggests that activity in the hippocampus and amygdala is lower on day 28 regardless of picture type. Figures 3, S3 and S4 also suggest very little effect of condition on the decreased activity and connectivity in the 28d group,

and there are no ANOVAs reported to provide evidence for an interaction between behavior, group and neural measure.

RESPONSE: It is correct that we do not find Group × Picture Type interactions in the univariate fMRI data, but a main effect of group and for some of the ROIs a main effect of picture type. However, it is not at all surprising that the 28-day interval has an effect on the overall activity for all picture types because a change in the memory representation over time implies that not only the old memories will be represented differently but also the process of separating these *old* pictures from *related* pictures and *novel* pictures will be different. However, the focus of the univariate analysis here is on disentangling the involvement of different areas in the memory task in general. We show here, that in the aHC the activity is reduced in the 28d-group compared to the 1d group in all conditions, suggesting that the contribution of the aHC to memory in general is reduced, while there is no reduction in activity in the pHC, suggesting that this area is still equally relevant for the task. We further show that the differential activity in these areas is functionally relevant, as it is differentially linked to indicators of memory specificity. Moreover, while univariate data only show overall activity in a ROI, multivariate patterns across voxels in an area carry important additional information. Therefore it could also be that an interaction between group (1d vs 28d) and picture type could be more (or only) present in analyses taking multivariate patterns into account. This was also what motivated us to run the representational similarity analyses (RSA) and indeed our model RSA data suggest that the similarity between *old* and *related* pictures changes in the 28d-group in comparison to the 1d-group. This indicates that the difference between the groups are also influenced by the picture type. We made this line of reasoning now more explicit in the manuscript. Please see page 6, lines 164 to 172:

„ Although we found a reduction of activity in the 28d-group in comparison to the 1d-group in the aHC and the mHC, it is important to note that there was no Group × Picture Type interaction (see Supplementary Figure 6) in these ROIs. Thus, our univariate results show that the activity in the aHC, but not the pHC, is reduced in the 28d-group compared to the 1d group for all picture types, suggesting that the contribution of the aHC in the task in general is reduced. Our brain-behavior correlations further show that the aHC, but not the pHC, is associated with memory specificity. It is not surprising that aHC activity was reduced irrespective of picture type after 28d because a specific memory representation is required to both correctly identify an *old* item as *old* and to correctly reject *novel* or *related* items.”

And on page 6, lines 177 to 183:

“While univariate analyses can show a general involvement of an area in a task, multivariate analysis allows the detection of specific patterns of activity across multiple voxels and may be more sensitive to the changing representations of the different picture types and more informative about the functional organization of memory at different time intervals. Therefore, we ran a representational similarity analysis (RSA, Fig. 4a) to examine whether the mnemonic representations differed in the aHC and pHC, whether they changed depending on the retention interval, and to what extent such different representational patterns can be linked to the proposed memory transformation.”

4. The authors state that this is ‘generally in line with the systems consolidation view, suggesting that hippocampal involvement in memory decreases over time’, the but none of the findings suggest a decrease that is specific to remembered or transformed items. Rather, there seems to be some other task-or group-level difference that is influencing all conditions.

RESPONSE: We agree that our univariate fMRI findings do not show a decrease that is specific to remembered or transformed items (please see also our responses to comments 3 and 7). However, the systems consolidation view does not predict the decrease in hippocampal activity to be

dependent on the level of transformation, or the item type (e.g. Squire et al. 1995). This theory states, that the hippocampus initially acts as a memory index for all (declarative) memories, that points to information stored in neocortical areas and binds them together to a coherent memory trace. Over time, i.e. with consolidation, the connections between the relevant neocortical areas strengthen till the hippocampus is not needed as an index for the retrieval of the memory anymore. This is assumed to occur largely in the same way for all memories; actually the distinction between transformed and detailed memories is not made in the classical systems consolidation theory. Thus, the systems consolidation theory expects a decrease in hippocampal contributions to the retrieval of all memories. We made this aspect explicit on page 5, lines 130 and 132:

“The fMRI findings so far are generally in line with the systems consolidation view, which would predict a decrease of hippocampal involvement in memory, irrespective of the specific picture type, over time^{8, 11, 23}”

5. Figure S5A, Table S2, and others: ANOVAs are not statistical valid means to test differences in FIR timecourses, as the time points are not independent of each other due to the inherent temporal correlations in BOLD signal. A more appropriate analysis would be to conduct statistics over the peak of the HRF (Schmitz et al, JNeuro, 2010; Turk-Browne et al., JNeuro, 2012) or conduct a permutation test.

RESPONSE: We agree and have now conducted the statistics over the peak of the HRF as suggested by the reviewer and described in Schmitz et al, JNeuro, 2010; Turk-Browne et al., JNeuro, 2012 and others. Importantly, our findings remained after this change in the analyses. Please see the respective changes in the results on page 5, the methods on pages 15, 16 and 17, in Figure 2 and Supplementary Figures 5, 6 and 7.

6. Figure 2C, 2D, S5B, and others: It is not valid to collapse across 1d and 28d groups in these scatterplots. The correlations are likely driven by the main effect of day on % of related false alarms and confidence, and the corresponding main effect of day on activity in anterior and mid hippocampus. Computing these correlations separately for 1d and 28d is the appropriate test of a correlation between BOLD activity and behavior.

RESPONSE: The brain-behavior correlations across all participants are in our view highly relevant because they show that the activity in aHC is indeed directly correlated with indicators of memory specificity, which cannot be taken for granted even if there are time-dependent effects on memory specificity and aHC activity. Please note that we have now also added statistics showing that the correlations with indicators of memory specificity are indeed reliably different for aHC and pHC (please see page 6, lines 152 to 154). Nevertheless, we completely agree that it is also interesting and important to look at the correlations for each group separately in order to rule out that the correlations are only driven by the described main effects. We found that in the left aHC the correlation between the percentage of *related* false alarms, the key indicator of memory transformation, and the ROI activity was significant in the 1d-group alone ($r = -0.45$, $p = 0.027$). For the 28d-group, this correlation did not reach significance ($r = -0.37$, $p = 0.121$), which might be *related* to the proposed decrease in aHC involvement in memory after 28 days. However, the strength of the correlations was largely similar and the lack of significance in the correlation in the 28 day group might also be due to a lack of power. For the memory confidence score, the correlations in the separate groups did not reach significance. We now report these additional analyses in the results section and added this information also to the legends of Figure 2. Please see on pages 5 and 6, lines 144 to 163.

“In order to further examine whether the decrease in aHC activity could be directly linked to the change in the nature of remembering, we correlated the activity in the

hippocampal subregions with behavioral indices of memory specificity, i.e. the FA rate for *related* lures and the Confidence Score. These analyses showed that specifically the aHC was associated with the specificity of memory. In particular, aHC activity for *related* pictures was correlated negatively with the FA rate to *related* pictures (left aHC: $t(46) = -5.15$, $p < 0.0001$, $r = -0.60$; Fig. 2c) and aHC activity for *old* pictures correlated positively with the Confidence Score (left aHC: $t(46) = 3.19$, $p = 0.0025$, $r = 0.43$; Fig. 2d). For the pHC, however, there were no such associations with memory specificity (left pHC = FA rate to *related* pictures: $t(46) = -1.08$, $p = 0.2879$, $r = -0.16$; Confidence Score: $t(46) = 0.44$, $p = 0.6645$, $r = 0.06$) and the correlations between activity and indicators of memory specificity were significantly distinct in the left aHC and pHC (FA *related*: $z = -3.46$, $p = 0.0005$; Confidence Score: $z = 2.81$, $p = 0.0049$). These correlations across the 1d- and 28d-groups indicate that the aHC and pHC are differentially linked to memory specificity. When we looked at the correlations separately in the 1d- and 28d-group, we obtained for the percentage of FA to *related* items, the key parameter of memory specificity, a significant correlation with aHC in the 1d-group only ($r = -0.45$, $p = 0.027$). For the 28d-group, this correlation did not reach significance ($r = -0.37$, $p = 0.121$), which might be *related* to the proposed reduced involvement of the aHC in memory in the 28d-group, although a lack of statistical power might also account for the non-significant correlation in the 28d-group. For the memory Confidence Score, the correlations with aHC activity did not reach significance in the separate groups (1d-group: $r = 0.24$, $p = 0.26$; 28d-group: $r = 0.21$, $p = 0.326$)."

7. If these statistical issues are appropriately addressed, the finding that anterior hippocampus exhibits decreased activity at 28d relative to 1d shows promise. However, stronger evidence for a time-dependent change in memory specificity would be to examine whether hippocampal involvement at 28d is decreases specifically for transformed items, and that such a decrease is not present for remembered items (or high-confident remembered items), or for forgotten items. As currently reported, there is no evidence that these effects are not a task- or group-level difference.

RESPONSE: The analysis suggested by the reviewer (i.e., looking at the differences between the 1d- and 28d-groups for transformed/forgotten pictures in comparison to the decrease for detailed (remembered) pictures) was not possible due to the expected performance differences between groups. In the 1d-group, memory performance was excellent and there were not enough transformed/forgotten trials to conduct a reliable analysis and therefore the critical comparison between groups would not have been possible. However, we did analyse the difference between groups for high confidence hits (vividly remembered pictures), and found a decrease in the 28d-group compared to the 1-d group in the whole hippocampus (and when dividing along the long axis, in the aHC) even for these high confidence hits. This again shows that the overall decrease in activity in the aHC does not depend on the picture type: there is less activity for low confidence hits than high confidence hits, but this difference is present in both groups, therefore there is no interaction effect. As stated above, this group-level effect in the univariate analysis is not at all surprising given that a memory representation is required for all picture types and a change in memory representation over time will therefore have an effect on the overall activity for all picture types. Influences of picture type were better captured in the RSA analysis and the changes in the representational patterns. We have added this aspect to the text. Please see page 6, lines 164 to 172:

„ Although we found a reduction of activity in the 28d-group in comparison to the 1d-group in the aHC and the mHC, it is important to note that there was no Group \times Picture Type interaction (see Supplementary Figure 6) in these ROIs. Thus, our univariate results show that the activity in the aHC, but not the pHC, is reduced in the 28d-group compared to the 1d group for all picture types, suggesting that the contribution of the aHC in the task in general is reduced. Our brain-behavior correlations further show that the aHC, but not the pHC, is associated with memory specificity. It is not surprising that aHC activity was reduced irrespective of picture type after 28d because a specific memory representation is required to both correctly identify a *old* item as *old* and to correctly reject *novel* or *related* items."

And on page 6, lines 177 to 183:

“While univariate analyses can show a general involvement of an area in a task, multivariate analysis allows the detection of specific patterns of activity across multiple voxels and may be more sensitive to the changing representations of the different picture types and more informative about the functional organization of memory at different time intervals. Therefore, we ran a representational similarity analysis (RSA, Fig. 4a) to examine whether the mnemonic representations differed in the aHC and pHC, whether they changed depending on the retention interval, and to what extent such different representational patterns can be linked to the proposed memory transformation.”

8. *Figure 4E: There no formal statistical test between model fits for the old/distinct versus old/related similar RDM. Without such a test, the interpretation that the anterior hippocampus best fits the old/distinct model and the posterior hippocampus best fits the old/related similar model is unfounded.*

RESPONSE: We compared the model fits now directly with directed paired t-tests. The results are reported on page 7, lines 208 to 218:

“Based on the behavioral data, we reasoned that the “*Old Distinct*” model should, in general, fit better in the 1d-group as these participants still had detailed memories, while the “*Old and Related Similar*” model might fit better in the 28d-group, as for these participants part of the memories had been transformed to gist-like versions. Our analyses showed that, in the 1d-group, the “*Old Distinct*” model had indeed the better fit in the left aHC ($t(23) = 1.86$, $p = .037$, one-tailed) and left mHC ($t(23) = 1.96$, $p = .031$, one-tailed), whereas in the left pHC both models were indistinguishable ($t(23) = -0.11$, $p = .542$, one-tailed). In the 28d-group, on the other hand, the “*Old and Related Similar*” model had a better fit in the left mHC ($t(23) = -2.61$, $p = .008$, one-tailed) and a trend towards a better fit in the left pHC ($t(23) = -1.42$, $p = .085$, one-tailed), while in the left aHC both models were indistinguishable ($t(23) = -0.32$, $p = .38$, one-tailed) and the model fits were generally rather low.”

Moreover, we went over the discussion section again to tone down our interpretations of the model RSA data, where appropriate. As one example, please see page 8, lines 251 to 252:

“...the model RSA data suggested that the representational patterns in the pHC resemble more gist-like patterns at the 28d retention interval.”

9. *It is unclear why left HC activation across the long axis is reported in Figure 2, but right HC activation is reported in Figure S5 even though the results are close to identical.*

RESPONSE: The pattern of results was overall very similar for the right and left hippocampal subregions. To reduce the number and complexity of the figures in the manuscript, we therefore decided to present only the data of the left hippocampus in the Figures of the main text and to present the data for the right hippocampus in the supplement. We have stated now explicitly in the text and in the Figure legends that the results are largely comparable for both hemispheres. Please see page 8, lines 219 to 220:

„The brain data was largely comparable for both hemispheres (see Supplementary Fig. 7 and 8 for results in the right HC)...“

10. *The reason for excluding 24 trials due to lack of data in a subset of participants is unclear. All data could be included with some changes to the analysis pipeline. RDMs could be generated using a weighted average to account for the number of trials per participant. Or, permutation tests could be developed to account for the different bin sizes across participants.*

RESPONSE: The toolbox we used for the RSA (rsatoolbox, Nili et al. 2014) requires exactly the same matrix size for all participants, as it stores the matrices in combined variables and uses them in functions that do not work with differently sized matrices. Thus, in order to be able to use this toolbox, we had to exclude 24 trials per participants, leaving 156 trials for each of the participants. The alternative strategies suggested by the reviewer are in our view more appropriate for MVPA but less for RSA. In particular, the RSA focusses on the representational patterns across a lot of distinct trials. If we would, for example, calculate the average per participant per picture category this would reduce the information space significantly. Furthermore, weighing the individual averages to account for the number of trials per participants would assume that trials across individuals are largely the same, which is unlikely to be the case given that there are considerable interindividual differences in memory performance and memory-related activity. In response to the reviewers comment, we explain now in more detail why it was necessary to exclude up to 24 trials per participant. Please see page 17, lines 519 to 527:

“Due to technical failure, we did not have functional data for all trials in some of the participants. Thus, in 24 trials data of one or more participants were missing. In order to later create average RDMs for each group (1d vs. 28d) and to use the other functions in the rsatoolbox⁴³, all single subject RDMs have to be of the same size and the same trial order, we therefore had to exclude these 24 trials from all participants, thus leaving the same 156 trials (52 *old*, 52 *related*, and 52 *novel*) for everyone. Although we exclude some data here, this procedure enables us to work with single trial activation patterns instead of averaging across picture categories, thus allowing for more information from distinct trials/pictures.”

11. *A discussion of similar papers showing time-dependent transformations of memories should be included. Both papers report distinctions along the hippocampal long axis.*

Ritchey, Maureen, Maria E. Montchal, Andrew P. Yonelinas, and Charan Ranganath. “Delay-Dependent Contributions of Medial Temporal Lobe Regions to Episodic Memory Retrieval.” *eLife* 4 (January 13, 2015): e05025. <https://doi.org/10.7554/eLife.05025>.

Tomparry, Alexa, and Lila Davachi. “Consolidation Promotes the Emergence of Representational Overlap in the Hippocampus and Medial Prefrontal Cortex.” *Neuron* 96, no. 1 (September 27, 2017): 228–241.e5. <https://doi.org/10.1016/j.neuron.2017.09.005>.

RESPONSE: The findings of these papers are indeed highly relevant in the context of the present study. In particular, both papers report some findings that are generally in line with our data: Ritchey et al. show that the aHC carried information for context in immediate and 1-d delayed memories, and although this is interpreted in a slightly different way in this paper, the reported memory representations for contextual details can be seen as rather detailed memory for the learned associations between objects and contexts. Tomparry & Davachi compared immediate to one week old memories and found data suggesting greater sensitivity to distinct features of memories in aHC and a bias toward representing overlapping information in pHC, therefore nicely in line with our data. However, our present data extends these studies in several important ways. In particular, the time interval of the present study is significantly longer than the interval in the previous studies (4 weeks vs 1 week or 1 day) and Tomparry & Davachi did not include measures of the proposed memory

transformation, as for them part of their stimuli where actually overlapping (different objects with the same scene) from the beginning, and the neural patterns for these overlapping memories were more similar in the pHC after a week. Thus, in this study there was actually no real behaviour indicator of the critical memory transformation process. In our study, the “only” overlapping part of our *old* and *related* stimuli is the (semantic) gist, with both different object details and scene details. Therefore, our study shows that memories were transformed to gist-like versions, and overlapping representations in the pHC can be associated with the gist. We added these relevant studies and their relation to the present study to our discussion on pages 8 and 9, lines 240 to 257:

“...In addition, our results fit to a study showing stronger aHC activity for recent memories than for remote memories¹⁹, a study showing that aHC carries information about memory contexts in immediate and recent (1 day old) memories²⁷ and studies showing a consistent implication of the aHC in memory for specific events²⁸. Our results further dovetail with reports showing that the aHC specifically is associated with segregating events²⁹ and with novelty detection^{16, 30}, both of which requires specific memory representations.

Whereas the activity of the aHC was reduced after 28d, no such decline was observed for the pHC, suggesting that not all parts of the hippocampus decrease in activity over time. The model RSA data, however, suggested a time-dependent change in the representational pattern in the pHC. While in the 1d-group the pHC representation was already less specific than the aHC representation, which corroborates the recent idea that there are complimentary learning systems within the hippocampus with one supporting gist-like representations^{32, 33}, the model RSA data suggested that the representational patterns in the pHC resemble more gist-like patterns at the 28d retention interval. This result is in line with a recent finding³¹, showing that neural patterns of overlapping memories were more similar in the pHC after a week of consolidation. In this study, however, part of the memories were actually overlapping (e.g. same scene with different objects), whereas our study extends this finding by having two different pictures with only the semantic gist as overlap, thereby pointing to a memory transformation process.”

Reviewer 3:

We thank the reviewer for his/her constructive and helpful comments. We are happy that he/she considers our report of interactions along the long hippocampal axis novel and interesting.

1. The authors conclude that "The present data point to a critical involvement of the aHC" in gist. Given that the data are correlational, I'm not sure how they would do so. At best they can indicate a "possible involvement".

RESPONSE: We agree and have changed the wording as suggested. Please see page 8, line 235:

"The present data point to a possible involvement of the aHC in the specificity of memory."

2. The authors use a temporal manipulation. When were the 1-day and 28-day scans gathered relative to one another? Were scans of each type intermixed to ensure there was no confound associated with changes to the scanner technical environment over time?

RESPONSE: Scans of each type were indeed intermixed, thus ruling out any systematic confound of testing delay and scanner environment. We have now added this information to the methods. Please see page 11, lines 322 to 324:

"The testing of the two groups was intermixed, so confounds related to changes in, for instance, the technical environment of the scanner over time cannot explain group differences."

3. The authors transformed each RDM using rank transform and scaling. The authors point out that this prevents comparison across regions, which is correct - these transformations also make it impossible to ascertain the degree of similarity observed at each time in each region, or even whether similarity values are just random distributions around zero. Higher rank values could also be expected for values that correspond to noise if similarities are in general negative, which is not unheard of. It's not clear why the authors would use these transforms - I suggest they use the pattern similarity r-values that they use in their computations for sake of transparency.

RESPONSE: The rank transformation and scaling was only used for the visualizations of the RDMs. This is the default option of visualizations in the rsatoolbox and is used to be able to better see patterns of similarities. Not rank-transformed RDMS would look like this:

It is in our view much easier to see the relevant information in the rank-transformed RDMs, as relevant differences in similarity values are made clearer. The information additionally available in these not rank-transformed RDMs (that similarities are, overall, much lower in anterior than posterior HC, with mid-portion HC in between) is what we showed in Figure 4c. We would therefore, if the reviewer agrees, prefer to use the rank-transformed RDMs for visualization. We added notes to the figure legends and to the main text to make it clear that rank-transformed RDMs are for the visualizations only. Please see page 7, lines 188 to 190:

“Note, however, that for the visualizations each RDM was separately rank transformed and scaled into [0, 1] preventing a direct descriptive comparison across hippocampal subregions. We therefore extracted the mean pattern similarity for each RDM: ...”

4. At various points the authors describe these rank-transformed values as e.g., “pHC memory representations”. Given how indirect the measures are, this is quite optimistic. It would be far more appropriate to describe the DVs as what they actually are, e.g., “pattern similarity values drawn from pHC”, as opposed to the construct the authors are hoping they measure.

RESPONSE: We completely agree and have therefore changed our wording at several locations accordingly. Please see, for example, page 8, lines 247 to 248:

“The model RSA data, however, suggested a time-dependent change of the representational pattern of the pHC.”

5. The authors argue that findings of more general memories, weakening hippocampal responses, and lower specificity of pattern similarity in pHC, are a clear indication of memory being transformed into gist representations. However, they do not explore reasonable theoretical alternatives. Can't this be explained more simply as a reduction in memory strength? This possibility should at least be discussed.

RESPONSE: We agree that the differentiation between memory strength and memory transformation is very important and this was indeed a central aspect of this study. In fact, we have designed the study explicitly to be able to differentiate between a reduction in general memory strength and memory transformation processes by the inclusion of the *related* pictures. If only memory strength is reduced after 28d, this would be reflected in a parallel increase in the FA rate for *related* pictures and the FAs for *novel* pictures. However, we found a marked interaction effect showing a much stronger increase of FAs for *related* pictures than for *novel* pictures. This strongly indicates that a transformation process has taken place, as participants after 28 days seem to remember the gist of the learned pictures and can still quite successfully keep that apart from the gist of entirely new pictures, but at the same time have problems remembering the details of the learned pictures, leading to a high FA rate for *related* pictures. Our brain-behavior correlations further show that the activity in the aHC, the hippocampal subarea that showed a time-dependent change in activity, was specifically correlated to the FA rate for *related* pictures (i.e., the key indicator of memory specificity/transformation). In addition, our model RSA data can also not be explained by a general reduction in memory strength, as this would mean that the similarity between *old* and *related* pictures should remain the same over time. We found, however, that the “*Old Distinct*” model could better explain the data in the 1-d group, whereas the “*Old and Related Similar*” model seemed to fit better in the mHC and pHC after 28 days. This strongly suggests that there was a decrease in the specificity/detailedness of the memories, whereas the gist memory was still intact, in line with the transformation view but at odds with the assumption of a general reduction in memory strength. We have now addressed this important point more explicitly and added a whole new paragraph to the discussion. Please see pages 9 and 10, lines 283 to 298:

“Finally, we would like to point out that the time-dependent changes reported here cannot be interpreted as a mere indication of a reduction in memory strength. In fact, we have designed the study explicitly to be able to differentiate between a general reduction in memory strength and memory transformation processes by including *related* pictures that allowed us to probe memory specificity. If only memory strength was reduced after 28d, this should be reflected in a comparable increase in the FA rates for *related* and *novel* pictures. We observed, however, a much stronger increase of FAs for *related* pictures than for *novel* pictures, which is in sharp contrast to the interpretation of a simple reduction in general memory strength but in line with the proposed transformation from detailed to gist-like memory. In addition, our model RSA data can also not be explained by a general reduction in memory strength, as this would imply that with time the memory for specific details and the gist memory decrease to a similar extent so that the relative representation of *old* and *related* items remains over time. Our data, however, show that the “*Old Distinct*” model best characterized activity in the aHC at a 1d-interval, whereas after 28d the two models were indistinguishable in the aHC and the “*Old and Related Similar*” model seemed to fit better in the mHC and pHC. Together, these findings indicate that, in addition to the well-known decline in memory strength over time, there is also a change in the nature of memory, from detailed to more gist-like.”

6. Likewise, alternatives to systems consolidation should be considered. The lack of reduction in cortical activity is not surprising given that it is a recognition test and stimuli are being directly shown to participants. The reduction in hippocampus activity could reflect improvement in memory, since the delayed stimulus presentation is essentially a repetition of the encoding phase, and better memory could correspond to more repetition suppression.

RESPONSE: Our behavioural data clearly show worse memory performance after 28d than after 1d. It is therefore in our view very difficult to argue that the reduction in hippocampal activity could reflect improvement in memory (as no such improvement is seen). We agree, however, that the lack of reduction in cortical activity is not surprising as stimuli are directly presented during recognition testing. We make this latter point more explicit on page 5, lines 127 to 130:

“As we tested memory with a recognition test in which participants directly view all pictures, it may not be surprising that neocortical areas were similarly involved in the 1d and 28d-group, as these areas might just reflect the processing of the currently viewed pictures”

7. Why were the behavioural data not tested for normality even though tests that require normality were employed?

RESPONSE: We have now tested for normality in the behavioral data, and indeed the Shapiro-Wilk normality test indicates that the data is not normally distributed for the hits and false alarms. However, the Confidence Score data is normally distributed. Despite this violation of the normality assumption, we have decided to still use the ANOVA tests because (i) the ANOVA is rather robust against violation of the normality assumption in big enough samples and (ii) the use of alternative, non-parametric tests would lead to a reduction of information (see, for example, Tabachnick & Fidell, *Using multivariate statistics*). We have now added this information to the methods section. Please see page 13, lines 397 to 400:

“The Shapiro-Wilk normality test was applied to the dependent variables: while the Confidence Score was normally distributed, this was not the case for the Hits and the FA. Despite this violation of the normality assumption we applied the above described ANOVAs due to the robustness of these tests against the violation of this assumption³⁷”

8. In figures, activity is described as a contrast value, but it is not clear what is being contrasted against in each case. This is important for interpretation and should be made clear in both the text and graphics.

RESPONSE: We agree and made this point clearer in Fig. 2 and Supplementary Fig. 7 and their figure legends. Moreover, we refer to the contrasts now also explicitly in the y-axes of the relevant figures.

9. I don't believe "(25, see also13)" is the right citation formatting.

RESPONSE: Thanks for noticing this mistake, which has now been corrected (please see page 8, line 238).

10. Fig. 4b is too small to be useful or see properly.

RESPONSE: Thanks for this hint. We have rearranged Figure 4, to make panel b bigger.

11. "consistent implication of the aHC in the recall of specific events^{19, 27.}" But the current study investigated recognition, not recall.

RESPONSE: It is correct that we test recognition. In this sentence, however, we wanted to point out that the aHC has been previously associated with memory for specific events. We have changed the wording to avoid confusion, please see page 8, lines 242 to 243:

“and studies showing a consistent implication of the aHC in memory of specific events”

Reviewers' comments:

Reviewer #1 (Remarks to the Author):

The authors have answered my previous comments satisfactorily and in detail. In particular, they have backed their results with appropriate statistics in cases where no such comparisons had been reported previously, and report results with a note of caution, when indicated. They interpret the comparisons of model fits in the different hippocampal subregions with more caution than other analyses (test here were only one-tailed, which is, however, justified given their a priori hypotheses), so I have no reservations concerning the conclusions drawn. I now recommend publication of their work.

I have only one remaining minor concern referring to point 2c) in the rebuttal:

COMMENT: l. 146ff, Fig. 3: "Specifically the right aHC-right amygdala connectivity for related pictures..." was an interaction analysis timepoint x picture type performed? Please report.

RESPONSE: In these analysis we did not perform FIR deconvolutions, but used the standard gPPI analysis in combination with Small Volume Correction. The results were significant for related pictures but not old or novel pictures in the analysis with the right aHC as seed. We now made this clearer in the results section, see page 5, lines 140 to 143:

"Moreover, connectivity analysis using the aHC and pHC as seed regions revealed that it was specifically the right aHC-right amygdala connectivity for related pictures (but not old or novel pictures) that was reduced in the 28d- relative to the 1d-group (SVC peak level: $x = 20$, $y = -4$, $z = -14$, $t = 3.59$; $p(\text{FWE}) = 0.0156$, $k=15$; Fig. 3)."

CONCERN: If no interaction between 28d/1d X seed region is reported, it is hard to draw definite conclusions about the specificity of results. A small additional note of caution might be appropriate.

Reviewer #2 (Remarks to the Author):

The revision was extremely responsive. The addition of the new analysis, separate correlations and comparison of correlation values, where appropriate, has now greatly increased the statistical presentation of the results.

One remaining concern is the issue with removing data for a rather minor technical reason. Specifically, 24 trials were removed in all participants because they were missing in only some. The reasons given for this are minor and fixable. The authors response to this was that the program being used does not allow for uneven bins. In this case (which comes up a lot in any RSA analysis that also includes a behavioral variable..) you can write your own code to compute the correlation between any two trials. The authors should include all data, whenever possible, to increase power. Thus, this one concern still needs to be addressed.

Other than that, I feel this paper makes a nice contribution to the literature.

Reviewer #3 (Remarks to the Author):

The authors have responded well to my comments. I have no further major concerns, although I request that they also include their numeric Shapiro-Wilk statistic, such that readers may themselves assess the degree of non-normality and whether the authors' use of an ANOVA was an appropriate choice.

Responses to reviewers

Reviewer 1:

We are glad Reviewer 1 was satisfied with our responses to his/her previous comments and that he/she recommends publication of our manuscript.

“Moreover, connectivity analysis using the aHC and pHC as seed regions revealed that it was specifically the right aHC-right amygdala connectivity for related pictures (but not old or novel pictures) that was reduced in the 28d- relative to the 1d-group (SVC peak level: $x = 20$, $y = -4$, $z = -14$, $t = 3.59$; $p(\text{FWE}) = 0.0156$, $k=15$; Fig. 3)”

CONCERN: If no interaction between 28d/1d X seed region is reported, it is hard to draw definite conclusions about the specificity of results. A small additional note of caution might be appropriate.

RESPONSE: We agree and therefore changed the wording accordingly in the results section, please see p. 5, lines 140 – 146:

“Moreover, connectivity analysis using the aHC and pHC as seed regions showed that the right aHC-right amygdala connectivity for related pictures (but not old or novel pictures) was significantly reduced in the 28d- relative to the 1d-group (SVC peak level: $x = 20$, $y = -4$, $z = -14$, $t = 3.59$; $p(\text{FWE}) = 0.0156$, $k=15$; Fig. 3), while we found no significant differences between the groups in the connectivity to the amygdala when using the pHC as seed region, suggesting that it might be the connectivity between the aHC and the amygdala that is notably reduced in the 28d-group.”

And the discussion section, please see p.9, line 281:

“However, this reduction in functional connectivity with the amygdala seemed to be specific to the aHC,...”

Reviewer 2:

We are glad Reviewer 2 found our previous revision extremely responsive.

One remaining concern is the issue with removing data for a rather minor technical reason. Specifically, 24 trials were removed in all participants because they were missing in only some. The reasons given for this are minor and fixable. The authors response to this was that the program being used does not allow for uneven bins. In this case (which comes up a lot in any RSA analysis that also includes a behavioral variable..) you can write your own code to compute the correlation between any two trials. The authors should include all data, whenever possible, to increase power. Thus, this one concern still needs to be addressed.

RESPONSE: As suggested by the reviewer in the following email correspondence, we have now created different sized RDMs for the participants using all available trials for each respective participant. Therefore no trials had to be excluded anymore. We report on see p. 17, lines 530-536 how many trials were available for participants for which not all trials were available:

“Due to technical failure, we did not have functional data for all trials in some of the participants. Thus, 8 participants had RDMs of slightly different sizes (in the 1d group three

participants had 179×179 RDMS; in the 28d group two participants had 179×179 RDMS, two participants had 178×178 RDMS and one participant had a 176×176 RDM).”

Using these new RDMS we could calculate the comparisons of overall pattern similarities across ROIs as before and we conducted the comparison with the model RDMS based on these new RDMS by creating model RDMS of the respective size matching the brain RDMS of each respective participant.

Thus, we now include all available trials whenever possible. However, the visualization of the RDMS and the direct comparison of RDMS between the 1d and 28d groups was not possible with these different sized RDMS. For visualization of the RDMS we created average RDMS for each group from the single subject RDMS and this average requires RDMS of the same size. And for the comparison between groups we aimed to directly compare the RDMS based on single trials, irrespective of picture category. To directly compare the trial-by-trial RDMS they had to be of exactly the same size. Therefore, we now use for the visualization and the group comparison shown in Figure 4d only data from the participants with all 180 trials. This is stated explicitly in the methods and the legend of Figure 4 and Supplementary Figure 8.

The pattern of results for these new analyses including all of the available trials yielded very similar results as the previously conducted analysis (including only 156 trials).

Please see the respective changes in the results part (p. 7-8, lines 194 – 222), the methods part (p. 17-18, lines 530- 567), Figure 4, Supplementary Figures 8 and 9 and the respective Figure Legends.

Reviewer 3:

We are glad Reviewer 3 has no more major concerns.

The authors have responded well to my comments. I have no further major concerns, although I request that they also include their numeric Shapiro-Wilk statistic, such that readers may themselves assess the degree of non-normality and whether the authors' use of an ANOVA was an appropriate choice.

RESPONSE: We agree and report the Shapiro-Wilk statistic now. Please see p 13, lines 402 to 406:

“The Shapiro-Wilk normality test was applied to the dependent variables: while the Confidence Score ($W = 0.96$, $p = 0.1246$) was normally distributed, this was not the case for the Hits ($W = 0.87$, $p = 0.0001$) and the FA ($W = 0.90$, $p = 0.0009$). Despite this violation of the normality assumption we applied the above described ANOVAs due to the robustness of these tests against the violation of this assumption³⁸.”

REVIEWERS' COMMENTS:

Reviewer #2 (Remarks to the Author):

The authors have now responded to my concerns about the analytic approach. I am happy to support publication.